# Animal health perceptions and challenges among smallholder farmers around Kaziranga National Park, Assam, India: A study using participatory epidemiological techniques

**Andy Hopker** [1] *, **Naveen Pandey**[2], **Sophie Hopker**[1], **Dibyajyoti Saikia**[2], **Jadumoni Goswami**[2], **Rebecca Marsland**[3], **Michael Thrusfield**[1], **Roopam Saikia**[2], **Sumanta Kundu**[2], **Neil Sargison**[1]

**1** Royal (Dick) School of Veterinary Studies, Easter Bush Veterinary Centre, University of Edinburgh, Roslin, Midlothian, Scotland, United Kingdom, **2** The Corbett Foundation, Village Bochagaon, Kaziranga, District Golaghat, Assam, India, **3** School of Social and Political Sciences, University of Edinburgh, Edinburgh, Scotland, United Kingdom

* ahopker@exseed.ed.ac.uk

## Abstract

Improvements to smallholder farming are essential to improvements in rural prosperity. Small farmers in the Kaziranga region of Assam operate mixed farming enterprises in a resource limited environment, which is subject to seasonal flooding. Participatory techniques, were used to elucidate the animal health challenges experienced in this landscape in order to inform and guide future animal health education and interventions. The flooding is essential for agricultural activities, but is a source of major losses and disruption. Farmers experience significant losses to their crops due to raiding by wild species such as elephants; predation of livestock by wild carnivores is also of concern. Access to veterinary services and medicines is limited by both financial and geographic constraints. Interviewees discussed nutritional and management issues such as poor availability of fodder and grazing land, while meeting attendees preferred to concentrate discussions on animal health issues. Livestock keepers were adept and consistent at describing disease syndromes. The key challenges identified by farmers were: foot-and-mouth disease; Newcastle disease; haemorrhagic septicaemia; chronic fasciolosis; diarrhoea; bloating diseases; goat pox; and sarcoptic mange. Improvements in the efficiency of farming in this region is a prerequisite for the local achievement of United Nations Sustainable development goals. There exist clear opportunities to increase productivity and prosperity among farmers in this region through a combination of vaccination programmes and planned animal management schemes, driven by a programme of participatory farmer education.

## Introduction

Smallholder farmers play an important role in global food production and engagement with them is an essential step in meeting United Nations Sustainable Development Goals. More

**Data Availability Statement:** All relevant data are within the manuscript and its Supporting Information files.

**Funding:** The Royal (Dick) School of Veterinary Studies provided support for this study in the form of staff time, travel costs, and subsistence in the field for AH. The British Biological Science Research Council (BBSRC) also provided partial funding in the form of a grant awarded to NS. Work at the Roslin Institute uses facilities funded by the BBSRC. No additional external funding was received for this study. The funders had no role in study design, data collection and analysis, decision to publish, or preparation of the manuscript.

**Competing interests:** The authors have declared that no competing interests exist.

than 80% of the world's poor people live in rural regions, and the majority of them are smallholder farmers [1]. Small farms provide food, employment, community, and sustainable land use, benefitting poor people in rural regions all over the world. Farms under two hectares are estimated to provide 30–34% of the world's food, while utilising only 24% of the world agricultural land, and as farms increase in size local biodiversity falls, and post- harvest food losses increase [2]. However, these farmers are often poor, and at risk of malnutrition, food insecurity and greater impoverishment [1].

Numerous factors can challenge the ability of smallholder farmers to run productive, profitable enterprises, these include: limited access to credit, seeds, markets and sufficient water; poor rural roads; and the impact of the climate crisis. At the same time there are great opportunities for development including: diversification of produce, empowerment of women, provision of rural employment, and improvements in prosperity and food security. Increased productivity in small scale livestock farming improves childhood nutrition. Increased agricultural productivity provides greater employment opportunities for landless labourers, as farmers move from undertaking daily labour themselves to employing labourers on their own holdings [3]. Small holder farming utilises very efficient value chains. Increased livestock farming activities by rural small holders, if supported through knowledge transfer, advisory services and co-operatives, lead to more widely increased economic activity in rural areas, better nutrition for both rural and urban dwellers, and where women are able to access these services equally, the empowerment of rural women, which in turn leads to better outcomes for children [4].

Smallholder farmers have an important role in improving rural prosperity in the state of Assam, India. There are an estimated 2.72 million operational agricultural holdings in Assam, of which 67.3% are under one hectare in size, the average size of holdings is just 1.1 hectare [5]. The population of Assam is 31.2 million people, of which 86% are considered rural [6], and 50% of the total population of the state are employed directly or indirectly in agriculture, with these jobs supporting a further 25% of the population as dependants [5]. However, despite this vast input of human effort, agriculture contributes just 19.34% of Gross State Domestic Product of Assam [7]. The reasons behind this low productivity are multifactorial: the use of traditional farming techniques; degradation of soils; poor access to credit; poor access to irrigation; and poor sales infrastructure have been identified as relevant factors [8]. Much of the food produced in Assam is consumed within the state, and while produce from livestock has increased since 1997, there is still a great shortfall compared with Indian nationally recommended food intakes [9].

The small size of agricultural holdings intrinsically raises challenges in achieving efficient production [5]; however, smallholder communities often have strong community cohesion, and farmers the importance of responsible stewardship of what is considered to be ancestral land. Small farm units have low levels of cash turnover which generates very limited funds for investment in farm infrastructure and modernisation. This low level of investment in turn necessitates continued high levels of time-consuming physical labour, which makes it difficult to generate opportunities and resources to undertake improvements, or for further education or training. However, as studies in other areas have found, a combination of training, modernisation, and investment can raise the productivity of small units [10].

Most smallholder farms in Assam are mixed cultivation and livestock units [5]. For the majority of Assamese smallholders, crop production is the main farming enterprise, but, the complementary role of livestock should not be underestimated. In Assam, the availability of grazing and fodder is strictly limited. Smallholders may be able to utilise rough grazing or free-access land; however, the quality of the pasture is often low, and land is frequently overstocked. Grazing areas are mostly confined to rough grazing, dry paddies, crop aftermaths, and road

edges. Dry fodder is usually straw and other by-products of rice and other crop production. Only 24% of farming households in Assam have any land at all used specifically for fodder production, and only 2% are using an area greater than 0.1 hectares for this purpose [11]. Genetic potential of animals combined with marginal diets limit productivity. The average milk production by a native cow in Assam is 1.91 litres per day and goats generally attain a slaughter weight of just 9.6 kg at eight months of age [11].

Assam is a humid sub-tropical monsoon region, Kopen climate classification Cwa [12], and much of the state is at risk of seasonal flooding; 39.58% of the land area of the state is classed as a flood-prone area [5]. This includes the area of study, which is subject to the seasonal flooding of the Brahmaputra River. Despite the local agricultural cycle being planned to take annual flooding into account, flooding results in damage to homes and property, disrupted livestock grazing, the deaths of animals, crop damage, and lost working days [13]. Animal movements, gathering of animals on high ground, and purchase of replacement livestock for those lost during flooding are key drivers of the spread of animal disease in the region [14].

Animal health is a prerequisite for efficient livestock production. Infectious disease and husbandry related conditions cause considerable mortality and impact negatively on growth rates, milk yields, reproductive performance, and available draft power. Animal keepers must devote time to the care of sick animals and veterinary medicines may not be affordable to all farmers. Provision of colostrum for neonatal calves in the region is frequently suboptimal, in part due to human consumption of bovine colostrum [15]. Infectious diseases known to be present in the region include foot-and-mouth disease, haemorrhagic septicaemia, Newcastle disease, classical swine fever, black quarter, anthrax and brucellosis [16]. Common known parasitic infestations include sarcoptic mange, fasciolosis, and gastrointestinal nematodes. Previous studies have examined the prevalence or impact of specific diseases such as foot- and-mouth disease [17]; haemorrhagic septicaemia [18]; peste de petits ruminants [19]; goat pox [14]; gastrointestinal parasitism and fascioliosis [20]. There is a pressing need to understand the relative impact of the full range of conditions on animal health and productivity in the region, and the effect of those conditions on the communities that depend on livestock for their livelihoods.

The achievement of United Nations Sustainable Development Goals [21] is hampered in this region by limitations to the productivity of farming, combined with the high input of human labour required. This is a barrier to increasing rural prosperity, educational attainment, equality, and social mobility. Sustainable methods are needed to empower smallholder farmers to improve livestock productivity, and rural prosperity. Access to agricultural extension education is variable in the region, and extension activities tend to be directed towards cultivation rather than livestock [8, 22]. Livestock immunisation programmes are frequently reactive in nature, vaccinating animals in the face of an outbreak. Planned animal management programmes have the potential to alleviate poverty [23]. The objectives of this research are to elucidate the important animal health challenges faced by smallholder farmers in a flood-prone rural region in Assam, and to identify areas where changes in practices or interventions may be effective with reference to the broader aim of achieving sustainable improvements to animal welfare and productivity. Here we report the use of participatory techniques to gain a deeper and more nuanced picture of the prevalence and impact of animal health conditions on farmers in the region.

## Materials and methods

### Study area

The region of study is located along a corridor of land bordered to the north by the Brahmaputra River and the Kaziranga National Park (KNP) and to the south by an area of highlands, the

Karbi Hills. The National Highway 37 (NH37) cuts through this land corridor, travelling from east to west (Fig 1).

Human entry into the KNP is strictly controlled through a tourist permit system, while all other penetration into the KNP is forbidden to prevent poaching and habitat degradation. This is monitored through a network of guard towers and Forest Guard patrols. Wild animals are freely able to leave the KNP and enter villages, leading to human- animal conflicts. The Brahmaputra River floods during the rainy season leading to seasonal inundation of villages, farmland, and forest, including the National Park. Typically, inhabitants of inundation-suscep-tible villages have to abandon their homes and farmland and move to islands of higher ground

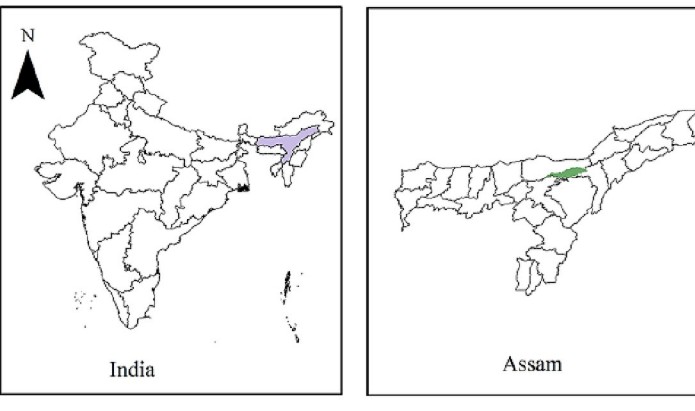

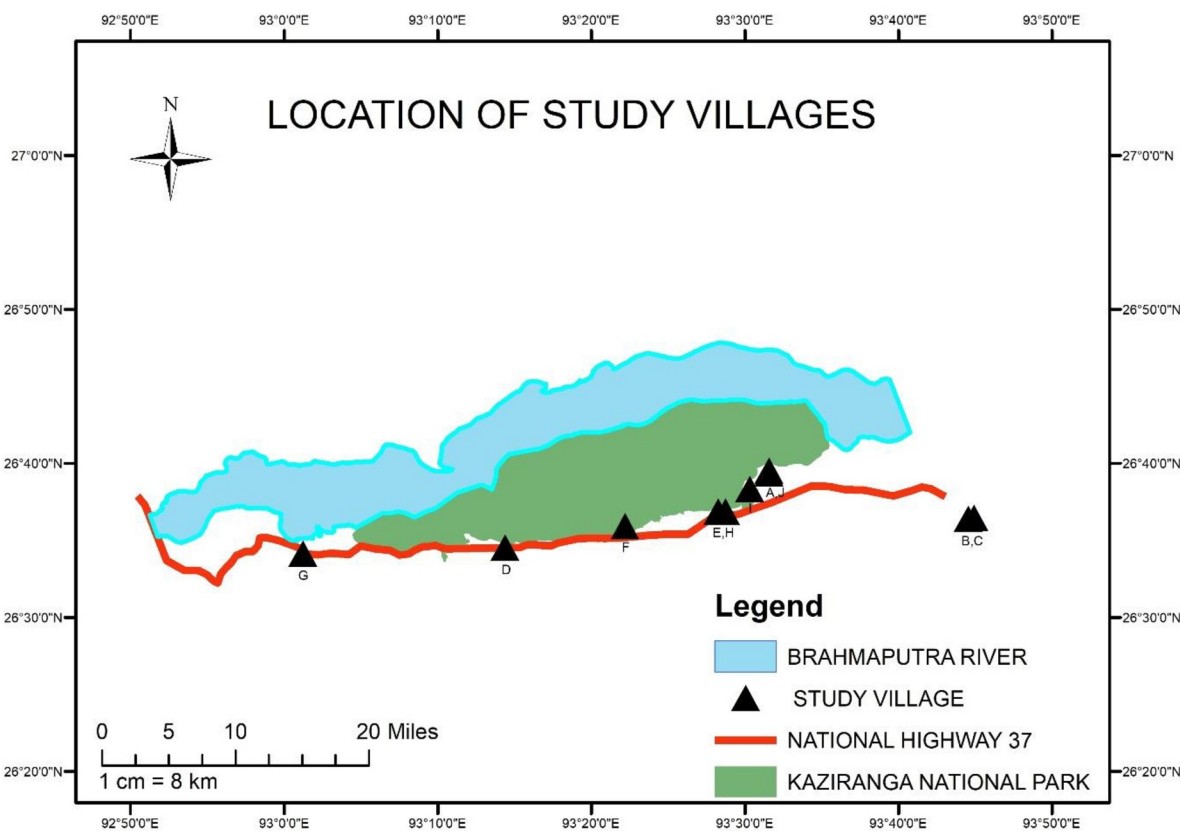

**Fig 1. Location of the study area and villages.**

for ten to twenty days annually, taking their livestock with them. This inundation also causes wild animals to leave the KNP, leading to a different pattern of human- wildlife conflict, including vehicle accidents, as these animals cross the National Highway 37 to access the high ground in the south.

Villages were purposively selected for participation to represent a range of land types, farming styles, and population. Six villages (A, E, F, H, I and J) were immediately adjacent to the KNP, and particularly susceptible to both flooding and human-wildlife conflict; two villages (D and G) were located close to NH37; and two villages (B and C) were located in the jungle on slightly higher ground, further from both the KNP and NH37. Village selection was made by local Corbett Foundation (TCF) community field workers with extensive experience in the region.

## Group meetings

Ten participatory village meetings on the subject of 'Animal Health' were carried out, one at each of the study villages. Meetings were open to all village inhabitants who wanted to attend and were advertised through word of mouth by community workers. Meetings were held outdoors at schoolhouses, prayer halls ('Namghar' in Assamese) and private homes. No village declined the offer to host an animal health meeting. Meetings were held at a date and time to suit the participants. The purpose, aims and scope of the meetings were explained to the participants at the start of the meetings, and attendees gave verbal agreement for participation, photography, and the use of results in publications. Signed consent and voice recording were not used as these are considered locally insensitive and inappropriate. Participants were informed that they could withdraw from the project if they wished, before asking if they wished to continue.

Following introductions and thanks, which included introducing the attending veterinary clinicians as "veterinary doctors", the discussion was initiated with the statement "We have all come here today to discuss the problems that limit how much your animals contribute to your household". An open discussion about livestock farming in the village followed, which then focused down to the identification of key animal health challenges by the participants. Once a number of key challenges were identified, ideally, but not strictly, limited to no more than eight conditions, these were discussed and described in greater detail by the participants. A proportional piling exercise was carried out on these key challenges, followed by a ranking exercise (see below). After completing the exploration of animal health challenges and data collection the meeting was then closed with a general discussion of potential mitigation and control strategies for these challenges using currently available resources, encouraging knowledge sharing between the farmers and drawing on the wider experiences of the facilitators. The facilitator endeavoured at all times to directly engage all participants to draw out the views of all attendees, and ensure the incorporation of the opinions of all participants regardless of age, gender, relative wealth, or social status.

## Proportional piling

Following the identification of key animal health challenges by the groups, proportional piling exercises were carried out, as previously described [24–26]. Each challenge was written in Assamese on a separate piece of paper, and these papers laid out on the ground. The group were then provided with 100 stones of roughly equal size and asked to distribute them between the key challenges to demonstrate the importance/ seriousness of the challenges, relative to each other. Participants were asked to take into account the severity of outcome, frequency of occurrence, duration of the problem, financial cost (including lost production, cost of

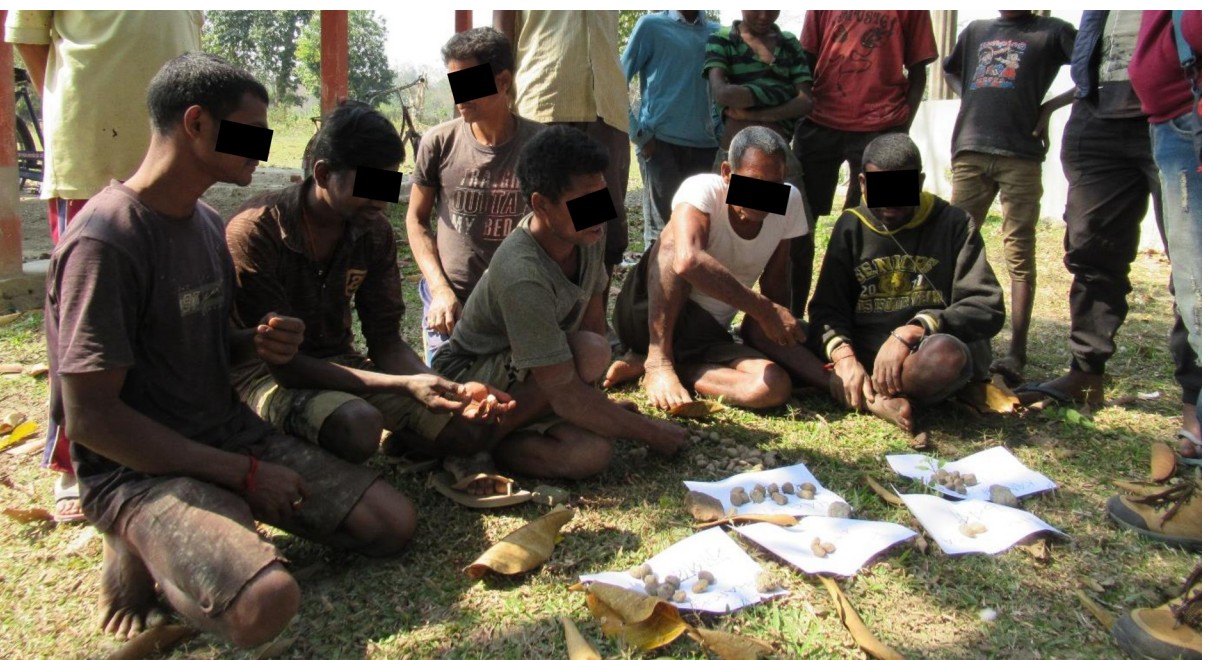

**Fig 2. A group of male farmers undertaking a proportional piling exercise discuss the distribution of stones to indicate the importance of various animal health conditions.**

medicines, and purchase of replacement animals), disruption to farming activities (including time of household members to care for animals or fetch fodder for those unable to graze and the loss of draught power), effect on human household nutrition, and emotional effect on the household.

Groups were free to arrange the distribution of the stones anyway that they liked. Some groups distributed the stones equally, or by the handful, to all members, for those members to add to the piles individually; or left the stones in a single pile and attendees were free to grab and distribute stones as they saw fit (Fig 2). Other groups nominated a few individuals to place the stones, while the remaining participants gave instructions, comments or advice on the distribution of the stones. Groups either nominated literate members as stone placers, or literate group members assisted non- readers. Once all the stones were all distributed the participants were then free to move the stones as many times as they liked until all the group was in agreement about the outcome (Fig 3). At this point, the facilitators counted the stones in each pile and marked down the score for each condition. The participants were then invited to redistribute the stones if they wished.

## Ranking and comparison

Following the completion of the piling exercise, in order to review the results of this activity with the participants, a comparative ranking exercise was carried out. The papers representing the animal health challenges were placed in a line in order of importance as previously described [26], either from top to bottom, or left to right, depending on the practicalities of the location. The facilitator then listed the names of the challenges in order and the participants were again invited to re-order the conditions if they wished. Groups were free to make no change, re-order the conditions only, or re-order the conditions and redistribute the stones if they wished.

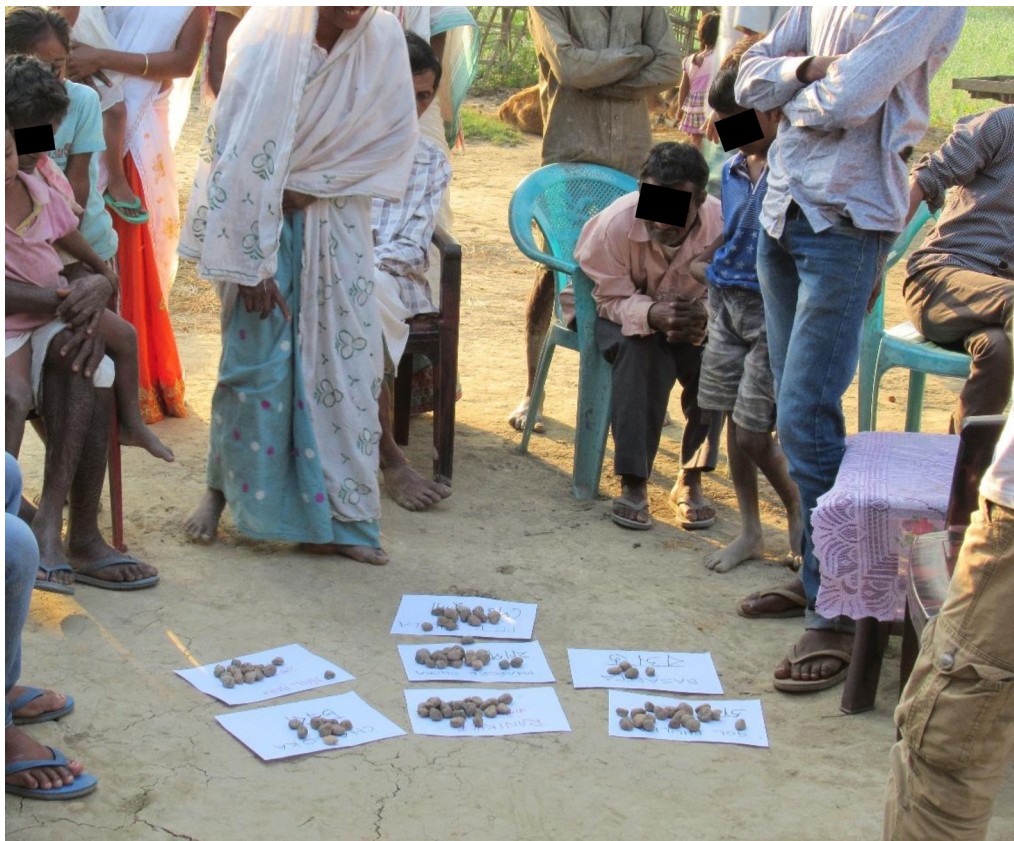

**Fig 3. During a proportional piling exercise, a female farmer re-distributes some of the stones laid down by her neighbours to reflect her opinions on the relative importance of various animal health conditions.**

## Village walks and direct clinical observation

Following each village meeting the facilitators would walk around the village with the group of farmers discussing farming strategies and techniques, animal health challenges and any other matters of relevance with the participants. Any sick animals would be examined at the request of the village inhabitants.

## In-depth interviewing

A series of interviews was carried out over two weeks in a cluster of three villages (A, I, and J) bordering the Kaziranga National Park. The data collection process was designed, and the results were analysed and reported, following the COREQ guidelines [27]. In-depth interviews were carried out at the homes of 17 smallholder farmers. Farmers were either interviewed alone, or together with one or two family members or friends. Additional family and friends frequently came and went during interviews, sometimes joining in for a time. Interviews were conducted in Assamese by a local bilingual (Assamese / English) Assamese community worker who was already known to the participants, and an experienced UK veterinary clinician. Each interview was arranged in advance with the help of community members. The researchers arrived at each household and waited to be invited in. All participants (farmers and researchers) typically sat in a loose circle at the same level. The community worker opened each interview with introductions, followed by description of the purpose and methods of the study, answered questions about the study, and offered the opportunity to withdraw, prior to taking

and recording informed verbal consent from the farmer participants. The interviews typically lasted two to three hours and employed open questions, with follow up questions as indicated, to create as informal and conversational an interaction as was realistic in the circumstances. The interviews covered farming practices, with particular reference to ruminant livestock keeping, and perceived challenges to health and productivity. Attitudes to farming, and the underlying socio-economic and environmental factors were also explored, including discussions about the role of education, alternative livelihood strategies, the effect of the proximity of the National Park, and the future of farming in the village. An *aide-mémoire* was used to ensure that similar areas were covered, and suggest relevant follow up questions where indicated. This can be found in (S1 File). The veterinary clinician posed each opening question, and then both researchers engaged in the ensuing discussion in a conversational manner. Participants were encouraged to freely discuss any and all aspects which were of importance or interest to them. All participants were volunteers, introduced to the project by a friend or relative, and care was taken to include animal keepers from a variety of social levels, families, and income levels within the village, and to ensure a balance of gender and age. No household declined to take part in the study. Prior to commencing the interviews, the aims and scope of the project, and possible uses of the data, including publication, were explained to the participants and verbal consent obtained. Written consent was not requested due to variable literacy between participants and a distrust of signed documents. Handwritten notes were made during interviews. Voice recording was not used because it was considered to be unacceptable to the participants, hence would have hindered the free exchange of ideas. Interview notes were transcribed and NVIVO 11 (QSR International Ltd) was used to build a node structure, to which responses were coded for analysis. The node structure comprised the following eight base nodes, with further sub classifications as required: Animal health education; Animal disease; Animal husbandry; Attitudes to/ effect of the KNP; Care of young animals; Description / opinions of the farm; Family information; Features of a good animal.

Interview participants were invited to attend village participatory meeting A. Meetings I and J, though in the same cluster of villages, were attended by different farmers. A further three meetings were held for the interviewees, to report the findings of the interviews, provide relevant animal health education, and advice on community animal health planning, in keeping with the aims of the study. Further relevant animal health education is planned for the other villages in the region, based on the findings of the study.

## Identification of conditions

The results of the study were examined by two Indian veterinary practitioners practicing in the locality and two UK veterinary practitioners with extensive experience of working on the Indian subcontinent. These clinicians evaluated and interpreted the conditions described by the farmers and agreed on the final definitions of the local terms. These conclusions were then further discussed with the respondents both to ensure accuracy and feedback the study findings to the participants.

## Ethical approval

Ethical oversight and approval were provided by the Royal (Dick) School of Veterinary Studies, the University of Edinburgh, Human Ethical Review Committee (approval number HERC 47 17). The aims and scope of the study were explained to the participants, and informed verbal consent recorded. The use of signed consent forms was not appropriate due to local mistrust of signed documents and varying levels of literacy.

## Results

### Attendance: Meetings, interviews

The ten participatory village meetings comprised an initial pilot meeting on 21$^{st}$ March 2017, three between 12$^{th}$ and 13$^{th}$ October 2017 and six between 12$^{th}$ and 17$^{th}$ February 2018. Three meetings were held in the morning and seven in the afternoon. Meetings were attended by groups of between 11 and 25 adults (mean 18 adults). A total of 184 adults participated in the meetings, 97 men (53%) and 87 women (47%). It is worth noting that morning meetings were less well attended by female participants. Meetings typically lasted two hours.

The 17 in–depth interviews encompassed 18 households, and were undertaken between 16$^{th}$ and 23$^{rd}$ March 2017. Interviews were carried out at the participants' homes. Eight interviews were with a single main respondent: 5 men and 3 women; nine interviews were with a pair of main respondents: 4 married couples, 2 daughters with their mothers, 2 sons with their mothers, and 1 pair of brothers. One of these interviews was with three main respondents, a married couple and a male neighbour representing a separate household. The stated age range of the main participants was 25–65 years (mean age 40 years). Additional family members or friends frequently joined interviews for a time. The group interviews, including the contributions of visitors, yielded interesting conversations and perspectives.

### Livelihood strategies

To properly evaluate the effect of various challenges to efficient production and to start to plan intervention strategies it is a pre-requisite to fully consider the people, their culture and livelihoods, and the landscape in which they live. Almost all inhabitants of the study villages identify themselves as farmers, though many pursue additional livelihood strategies. These include professional employment, such as local teachers; service jobs, particularly in the tourist industry (including hotel managers), community workers, and drivers. Trades include carpenters and seamstresses; and engagement in business activities such as procurement and sale of goods. Weaving is a common method for women to access additional income. Some inhabitants keep small shops. Daily labour is also undertaken by some farmers to earn additional cash income. A table showing household income streams of interviewees can be found in the (S1 Table). Respondents engaging in farming as their sole income stream were proud of this, and those engaging in multiple livelihood strategies were equally proud. The majority of village inhabitants had attended primary school, though there are still illiterate people in the villages, particularly older people. In modern times most children attend high school, though not all graduate. Very few children from the villages progress to University. All interviewees expressed a desire that their children work hard at school so that they are able to access better jobs and living standards; a few also expressed sadness that if their children attained this then they would probably move away from the village for work.

### Animals kept and their uses

Knowledge of the livestock species kept, and the uses of these animals is essential both for planning farm strategies which maximise the efficient use of resources, and for considering disease control strategies. All of the ten villages kept cattle, goats, chickens and ducks. Eight villages also kept pigeons, seven kept draft oxen, six kept pigs and five kept buffalo. Participants describe keeping cows for milk to drink and sell, to produce calves, and using dung to fertilise crops and vegetables. Goats are usually homebred and are kept for sale only, particularly at times of financial need or hardship, and are considered an important form of savings. Few participants report eating their goats, and then only at a time of celebration, such as a family

wedding. Goats are not commonly milked in this area. Chickens and ducks are kept for sale, household meat consumption, and eggs for home consumption or sale. Ducks are generally considered more valuable than chickens. Pigeons are generally kept for home consumption only. Pigeon meat or eggs may be consumed by the household, but not always both. Draft oxen are used for cultivation, drawing carts, threshing crops and a variety of other traction tasks. Their dung is used as fertiliser. In some villages the use of mechanical cultivators and tractors had entirely replaced oxen, however many respondents still consider oxen to be superior due to versatility to provide power for other activities; the flexibility to cultivate when they want, rather than according to a machine operators schedule; and the quality of the results of cultivation. Oxen which have been trained for draught work are valuable for sale, and training bullocks for this purpose can provide additional income for those farmers with the necessary skills and time available. Pigs are kept for the same purpose as goats, and may be home bred or bought in as piglets. Buffalo are kept for milk or draft work, depending on their type, as well as providing offspring for sale, and dung as fertiliser.

## Features of a good animal

Informed knowledge of the local concept of a 'good' animal provides insight into animal health priorities, animal health knowledge, and aims of animal production. It is also an indicator of the ability of farmers to recognise ill health. The selection and purchase of a cow or oxen is a predominantly male activity. Generally, a utilitarian view was taken to the selection of livestock. Large body size, provided it was in proportion, was stated by fifteen out of eighteen interviewees as a good sign for milk production or draught work. Udder size, teat conformation and large milk veins were considered positive factors for milk yield and ease of milking, mentioned by ten interviewees. A large belly and rump were considered advantageous for the production, birthing and care of strong, healthy calves. A large bellied cow was considered to have a better appetite, and this increased food consumption would lead to improved milk yields and greater calf birth size. Height was considered to be a secondary advantage in cows, but overall body size was more important.

> "For oxen, the size of the thighs is most important. For a cow her length, belly size, and udder size matter most. A plump udder is good, a flat one has no milk. The look (prettiness) of the animal is not important."
>
> Interviewee 1, male, 49.

The size and physique of an ox were considered particularly important, especially the musculature of limbs. A large hump was considered an indicator of health, stamina and strength; however, it was stated that it was important that the shape of the hump did not impede the position of the yoke, leading to hump sores. Increased height was a positive factor in draught oxen, and long legs on young oxen, were considered a sign for further growth potential. One participant describes the selection of oxen at market:

> "Check his teeth, this also tells you his age. Look at his physical figure, then touch the body and feel his power."
>
> Interviewee 5, male, 39

Farmers were alert to the signs of health, and the absence of signs of ill health in animals. Locomotion, posture and hoof conformation were considered important in cows and

particularly oxen, as was seeing the animal display a good appetite. Bright eyes and erect ears were also considered signs of good health. A female farmer describes desirable attributes of cattle:

> *"Colour, health, physique. Ears erect, eyes. You know if an animal is healthy by looking, legs are muscular, the hoof should be pointy, tight shape, upright, with a small gap (between the toes). . . Colour- red and black mix, no white. White is not so good, not suitable for this home. . . Cows should have a strong rump, if the cow and calf are together, then look at the health of the calf. . . To buy animals (my father in laws') sons all go together. Sometimes an experienced man from the village goes with them if available."*

Interviewee 18, female 22

Five male interviewees said that they would have an experienced or knowledgeable man from the village accompany them when they went to purchase a cow or oxen. It seemed that this service would be freely offered by a friend or neighbour, however the man receiving advice might, for instance, provide lunch for his neighbour.

> *"The animal's legs, walking style, health and body size are all important, but I don't have that knowledge. I only look at its legs and its walking style and I guess if it is a good animal. I take an experienced man from the village with me when I go to buy cattle, and I listen to what he says."*

Interviewee 11, male, 58

Two female interviewees described their husbands as knowledgeable men whose help might be sought in the selection of new livestock. Two respondents discussed the importance of questioning the vendor about the milk production of the cow or the cow's mother, while another said that it was important to see a cow's previous calves if possible.

Other considerations included hair colour, though participants were careful to point out that this was a secondary factor. The preferred (or avoided colour) varied between houses (or households?) within the village.

> *"Colour (of the animal) should suit the house. Black and red are right for this house. Colour is not the main consideration though, other things must come first."*

Interviewee 4, male, 27.

The animal's eyes, facial structure, hair quality and horn shape (or absence) were also considered by some respondents, but only as secondary factors. A long tail was also considered an advantage to keep off flies. Six interviewees mentioned aesthetic considerations such as colour or facial features, however four of these interviewees stressed that these were of secondary importance only.

## Farming, environmental challenges, conflict with wildlife, and seasonal flooding

The primary crop cultivated in the area is rice. Traditionally farmers are dependent on the seasonal inundation to irrigate their land and flood their paddies. While essential to the local agricultural and environmental cycle, the seasonal flood also causes major disruption, damage to property, and mortality of animals every year. All 18 interviewed households discussed the

seasonal flood, and three households stated that it was the greatest barrier to agricultural productivity.

> "*My dream is to buy land on high ground that won't flood. Then I won't have to take my cattle to the road and live there during the flood.*"

Interviewee 1. Male, 45.

Lack of sufficient fodder for livestock was raised as a major barrier to productivity by six households.

> "*Less availability of grass. . . I am unable to buy good quality food for my animals. We don't have that kind of money so we are unable to provide good nutrition. Sometimes I buy food, but not enough for all my animals, and this reduces production. Husk and grain that I mix in cooked food. . . Animal nutrition is the biggest problem from March to July.*"

Interviewee 11. Male, 58

The problem of animal fodder becomes particularly acute from the later part of the dry season as plant growth slows, and then during the seasonal flood grazing land is covered, and providing adequate nutrition for livestock at this time is especially problematic and a limit to productivity.

> "*(there is) no grazing during the flood. We give banana tree chopped, elephant grass, rice straw, and weeds from the wetland.*"

Interviewee 14. Female, 34

Factors occurring around the seasonal flood, including animal movements and environmental conditions contribute to outbreaks of infectious disease every year.

> "*Chaboka, dhoka- dingra, bhekulia. They all occur after the flood*"

Interviewee 3. Female, 30

Most smallholder farmers grow just one seasonal rice crop per year, but another, unseasonal rice crop, is also possible for those able to purchase and fuel a diesel- powered irrigation pump. Some farmers may elect to grow an unseasonal crop in preference to a seasonal crop to reduce crop-raiding by wild animals. Farmers report losing between 20% and 50% of their annual rice crop as a result of crop-raiding by wild animals, primarily elephants. However wild animals quickly adapt, and patterns of crop-raiding rapidly shift, allowing animals to exploit the unseasonal crop. The issue of crop raiding was raised by 14 households from the 18 interviewed.

> "*Usually we are doing crops in the winter time. The flood washes out every crop. Now we are doing the (*un*)seasonal crop because we have the water motors* (pumps) *and things are improving bit by bit. Because of the flood and the crop raiding pattern, we changed. Now the raiding pattern is changing too. . . If not raided- 3 bighas is enough to support us. I get 4–5 quintals (1 quintal = 100kg) per bigha because I use less fertiliser and pesticide. Elephants take more than 50% of our crop. From the 2016 seasonal crop I got only 200kg from 4 bighas due to elephant raiding.*"

Interviewee 16. Male, 52

The bigha is a measurement of land area. The size of a bigha varies in different regions of India. In Assam 1 bigha is equal to 1340 $m^2$, approximately 0.33 of an acre.

Farmers commonly sit up all night to protect their crops, either in tongis (watchtowers) or sometimes on the ground. Recently The Corbett Foundation (TCF) constructed solar powered electric fences in some areas to assist in protecting farms from crop raiding. TCF have also funded the construction of additional tongis at strategic locations, provided solar powered torches, and helped organise rotas for guarding crops.

*"Yes, crop raiding is less since the* (solar powered) *fence was built* (by The Corbett Foundation). *The fence is helpful. Crop raiding is less this year. Our main problem is the flood. Now we are cropping in a different season, but this means we are increasing spending on water."*

Interviewee 1. Male, 49

*"We mostly rely on crops. Crop raiding by elephants and wild boar is our most important problem. This is the main problem and nothing else. Sometimes elephants eat banana trees. It is in their nature, it is not a big problem. Forest guards do come to help us with this problem. TCF have helped us providing tungi and torches. These have helped us to scare off elephants, we are very grateful to them."*

Interviewee 17. Male, 55

Farmers, particularly those working alone, must dedicate a great deal of time and energy to protecting their crops overnight in addition to working their farms and attending to other employments and household activities during the day. Solar fences, in areas where they are present do mitigate crop raiding, but they do not stop it.

*"Protecting crops from a tongi* (watchtower) *is hard work and I am up all night. The solar fence does not protect us from crop raiding. I have no cultivation land of my own. I work another man's land and give him 50% of the crop. Crop raiding is a village problem."*

Interviewee 2. Male, 45

Most discussion of crop raiding was by male farmers; this is explained because guarding crops overnight is customarily a male activity. Guarding crops is not only tiring, it also carries a risk of sustaining serious physical harm, due to the inherent risks in driving off wild elephants and other species of megafauna.

A secondary crop, usually mustard seed, or sometimes pulses (black daal) is also grown. These are cultivated on the dry paddy fields following the harvest of the rice crop. Vegetables are primarily grown for household consumption; however, surpluses are sold, and some farmers may cultivate vegetable as an additional crop for sale. Farmers typically own only a few acres of land, sometimes less, though they may access more land for cultivation by renting from the government, private rental, or the system of "adhi" where the land is farmed by a person other than the owner, and the profit is split equally between the owner and the farmer. The adhi system may also be applied to livestock, whereby the owner of the livestock gives it into another person's care, and the owner and the carer split the profit of any sales.

Another type of human- wildlife conflict occurring in the region is predation of livestock by large carnivores, typically tigers in this landscape. While more sporadic, the financial and emotional burden of each loss is considerable, and is a major source of worry to farmers. Eight

interviewees discussed the problem of predation of livestock, six of whom related incidents when their animals had been killed. Two interviewees discussed predation as a major worry, but stated that their own animals had not been killed.

> *"We are poor people raising cows and ox. Our biggest problem here is the tiger. Tiger causes a lot of damage. If it kills a cow or an ox worth Rs 30,000 how can we deal with that. We need help from Forest Department or some NGOs to deal with this problem."*

Interviewee 11. Male, 58

Eight interviewees discussed wanting to purchase more highly bred animals, indicating that they consider the genetic potential of their livestock to be a limit to production. Four interviewees linked the acquisition of higher yielding animals with the need to secure more fodder resources in order to feed these animals to meet their potential.

> "(I would) buy high quality hybrid cows to increase production. I think I can produce for most of the year, but there will not be sufficient fodder in flood time. That is why I am not buying (cows)."

Interviewee 18. Female, 25.

Two interviewees specifically mentioned money as a barrier to farm improvement, however 8 more suggested changes that they are currently unable to make which would require capital investment. Farmers perceive the barriers to farm improvement as complex and interlinked, with an interplay of lack of financial resources, wildlife conflict, and the seasonal flood.

> *"Because of flood, this ground is not high enough. I want to buy improved quality cows but because of the flood problem I am not buying. I want to make this ground higher, then perhaps I will buy. I want to make a good shed for the cows and then I can buy good quality cows and make more milk. I want to make a good shed with some metals* (steel mesh) *and then I can protect them from the tiger."*

Interviewee 11. Male, 58.

## Animal health challenges

Identification of key challenges to animal health and productivity is central to planning effective interventions. Three methods: proportional piling, ranking, and interview data were used to elucidate these. Collation and comparison of disease descriptions from interviews and meetings, followed by assessment by four experienced veterinary clinicians was used to ensure the accuracy of disease identification.

**Interpretation of participants' descriptions of conditions.**   The identification of animal health conditions from the local Assamese terms, using descriptions gathered from meeting groups and individual interviews, as well as clinical observation, as agreed by the team of veterinary clinicians, is shown in Table 1, along with the frequency of ranking (by meeting groups) or discussing (interviewees), and the total proportional piling score from all meetings combined. The descriptors of disease and the frequency of their use by meeting groups and interviewees is shown in supplemental material along with relevant testimonials (S2 File), and these descriptions are used, with clinical observations, to identify the conditions.

Reviewing these conditions considered of particular importance by participants, it can be seen that seven Assamese terms encompass four specific infectious disease plagues, three

**Table 1. Likely diagnoses of locally named conditions.**

| Local condition | Likely diagnoses | Number of times condition was: | | Total proportional piling score (10 meetings, 1000 stones) |
| --- | --- | --- | --- | --- |
| | | Ranked by meeting (n = 10) | Raised by interviewee (n = 18) | |
| **Chaboka / Kurra phata** | **Foot and Mouth Disease** | **10** | **17** | **133** |
| **Ranikhet / Murgi-julka- loga / Hal- julka-loga** | **Newcastle disease (Differential diagnosis: Avian influenza)** | **9** | **3** | **122** |
| **Gol phulla / bhekulia / (Dhoka dingra, dingi phulla)** | **Swollen neck, chin or throat: Haemorrhagic septicaemia and/ or Chronic fluke** | **8** | **10** | **127** |
| **Hagoni** | **Diarrhoea of goats. Participants describe a condition similar to *Clostridium Perfringins* enterotoxaemia. A mass hatch endoparasitic event could also be possible** | **8** | **10** | **112** |
| **Pet phulla** | **Bloat** | **7** | **13** | **62** |
| **Sagolay bohonta** | **Goat pox** | **7** | **-** | **113** |
| **Papora dhora** | **Sarcoptic mange on goats** | **6** | **8** | **96** |
| Letekua | Corneal ulcer and/or keratoconjunctivitis | 2 | 6 | 15 |
| Jor | Fever | 2 | 3 | 26 |
| Moh- bis- oni | Black quarter | 2 | 2 | 15 |
| Kandh Singha | Hump sore | 2 | 2 | 29 |
| Murigoni | (Listerial?) meningioencephalitis | 2 | 2 | 7 |
| Sikora | Ticks | 2 | 3 | 19 |
| Pelu | Gastro- intestinal worms | 2 | 1 | 25 |
| Bohonta / basanta | Pox /dermatitis (non- specific) | 1 | 8 | 4 |
| Anthrax | Anthrax | 1 | 1 | 6 |
| Pani howa | Conjunctivitis? | 1 | - | 10 |
| Goat hoof cracks | FMD? Contagious Ovine Digital Dermatitis? | 1 | - | 14 |
| Hafonee | Impaction? | 1 | - | 13 |
| Geva- ghuti (Dhenia) | Tongue disease? | 1 | - | - |
| Bat- hera | Mastitis of goats (Agalactic- possibly Caprine Arthritis Encephalitis or Mycoplasmal in origin?) | 1 | - | 5 |
| Poor growth goat | | 1 | - | 4 |
| Okoni | Fleas and/ or lice | - | - | - |

Notes: Denotes that the condition was either not ranked by any meeting group, not discussed by any interview participant, or not included in the piling exercise by the participants.

Ranikhet: Newcastle disease appears the most likely diagnosis as the primary signs were neurological, particularly "sleepiness", dung was reported to be white in colour, and ocular signs were not reported by participants. However, Avian influenza is an important differential diagnosis for this condition.

parasitic diseases, and three primary clinical signs which can encompass a variety of aetiologies.

**Proportional piling by village farmer meeting groups.** Fig 4 shows the results of the proportional piling exercise, by village and condition. Fig 5 shows the combined results of the piling exercise for all ten villages. These are the final results, as agreed by the participants following discussion and re-piling. As discussions were open, conditions were included in the piling and ranking exercises by the participants' suggestions only. Therefore, not every group discussed or ranked the same exact same group of conditions.

**Ranking of conditions from 1–8.** Using the results of the piling exercise, conditions were ranked in order of importance from 1–8 (one highest, eight lowest). On inspecting the order

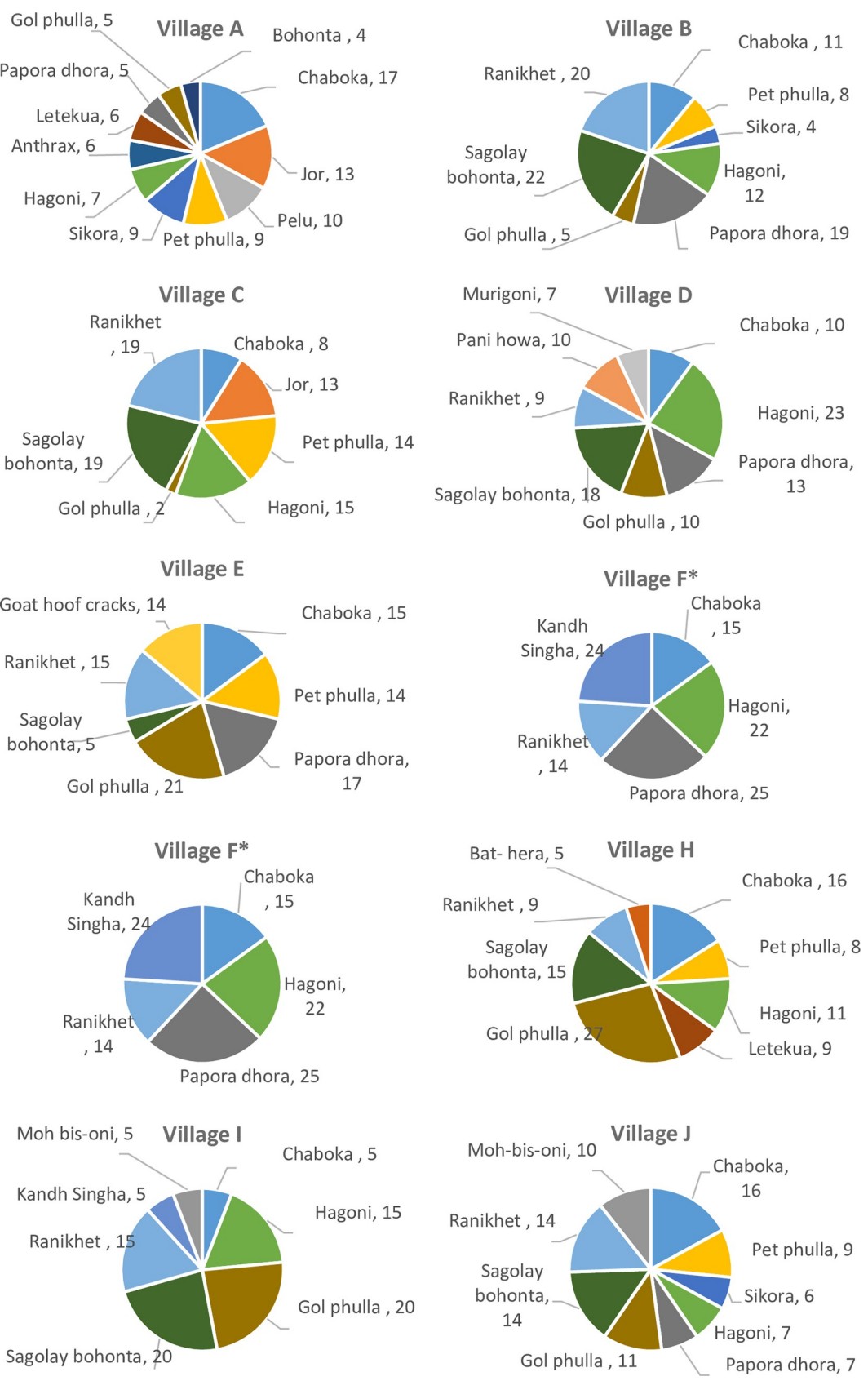

**Fig 4. Pie charts showing results of proportional piling exercises to indicate the perceived relative importance of animal health conditions for each village A-J.** Notes: Village C wanted to rank conditions initially, then re-ordered them following the piling exercise. Village D discussed Ranikhet, a disease of chicken and ducks, at length during the open discussion, but did not want to include Ranikhet in the proportional piling exercise as they identified themselves as cattle and goat farmers. However, when asked to rank the conditions considering the financial impact, they placed *sagolay bohonta* in position 1 and Ranikhet in position 2. Conditions were re-ranked, but piling results were not altered by the group. Village F: Undertook the piling exercise based on prevalence, then when they saw the results, completely re-ranked based on disease impact, but did not wish to repeat piling exercise. Village H: Undertook the piling exercise, discussed the results, then re-did the piling exercise from the start.

of importance participants were free to re-rank as they chose. The results of this ranking exercise are shown in Table 1. Fig 6 shows the all the conditions identified as of key importance, by meeting groups and interviewees.

Comprehensive details of village meetings, attendance, animal species kept, disease descriptors, breakdown of proportional piling exercises, and interview notes can be found in (S3 File).

## Animal health education

All eighteen interviewed households expressed a desire for animal health and husbandry education to be available for them or their children, and all expressed a preference for face to face practical training, eleven also expressed a role for printed materials, posters, books, or leaflets.

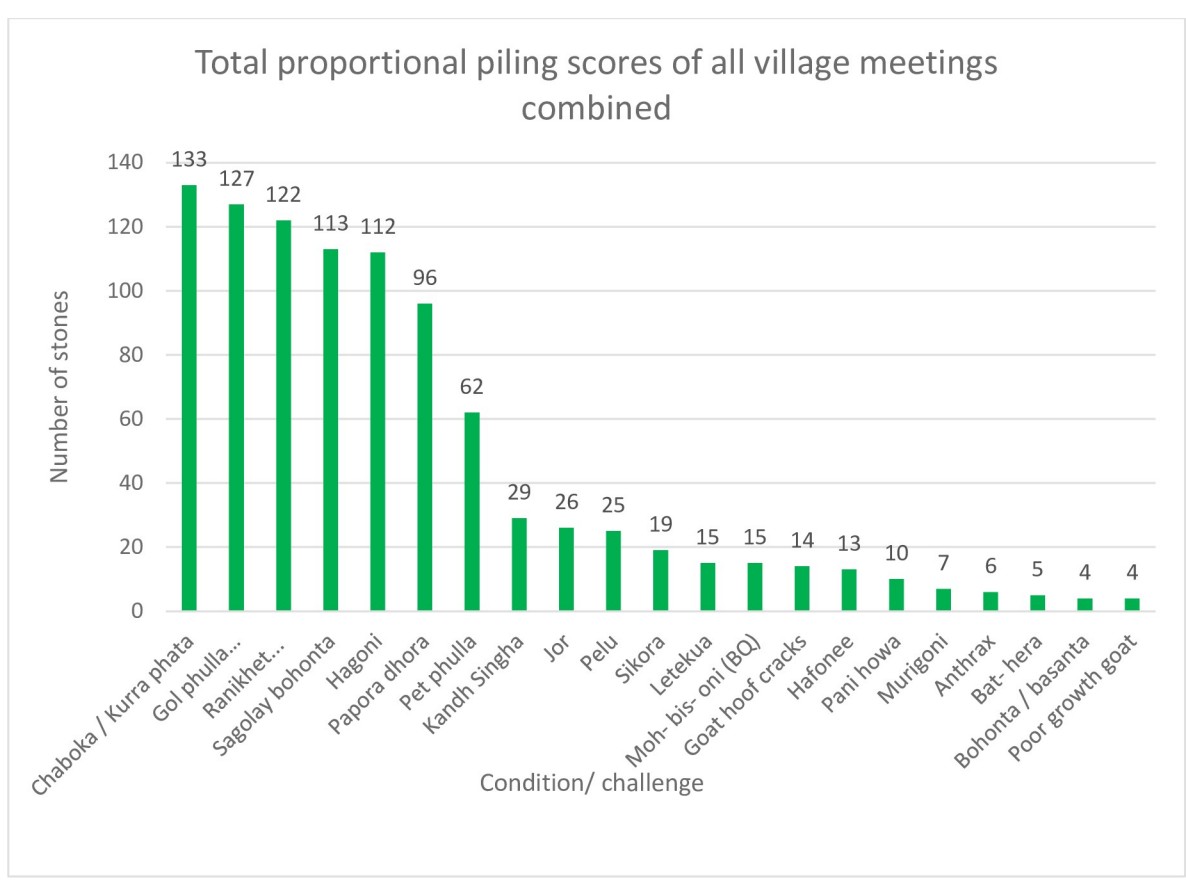

**Fig 5. Total of proportion piling scores to indicate the perceived relative importance of animal health conditions for all ten villages combined.** Total number of stones used is 1000.

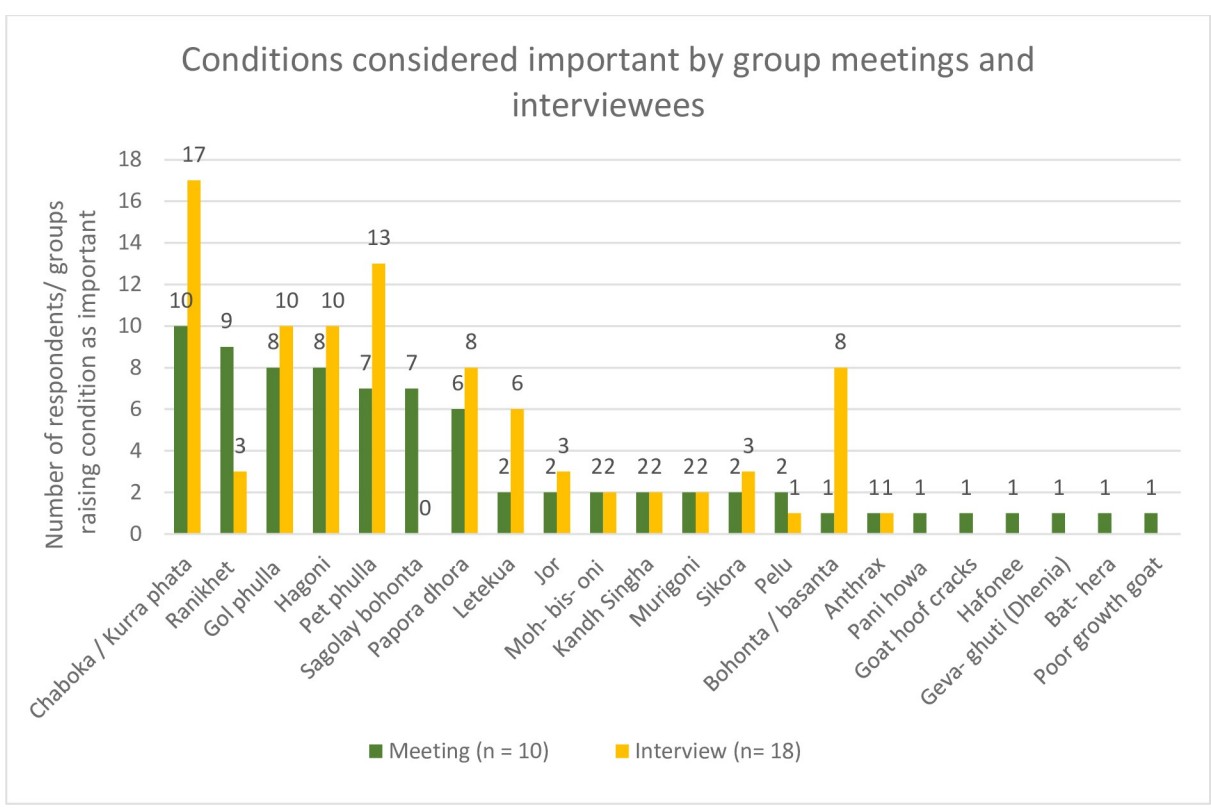

**Fig 6. Frequency of animal health challenges considered of importance by meeting groups (n = 10) and raised for discussion by interview participants (n = 18).**

*"I can read books but some things can be learned only from practical training. Posters and leaflets are also helpful. (What things might make it difficult for you to learn?) Lack of resources, the flood. I have to work hard to send my boys to school so time is limited."*

Interviewee 13. Male, 52.

Three interviewees stressed the importance of printed materials being in Assamese (not Hindi or English), while five interviewees cited difficulty reading Assamese as a barrier to learning. Seven interviewees raised the issue of finding time to learn, either as lack of time being a barrier, or the need to make time as a necessity, two stated that learning would be particularly difficult during the flood season. Six interviewees cited lack of resources as a barrier to learning, three a lack of an educational environment on the village, and three people cited a lack of previous education. Two people specifically stated that any training must be delivered in the village.

*"Training selected people would be more helpful. We lack the proper resources for learning here. Teaching from someone with experience about animal health and farming would be the best way for people to learn and apply knowledge. (Reading materials?) Yes, people can learn from them. A few might be interested, but not all. People here are too busy cultivating food. "The literacy rate here is about 30% (*for what language? Assamese?*). Yes, Assamese. A few people here can read Hindi, and English too. The young people read better, the old people less."*

Interviewee 8. Female, 25 (Primary school teacher)

Comprehensive details of village meetings, attendance, animal species kept, disease descriptors, breakdown of proportional piling exercises, and interview notes can be found in (S3 File).

## Discussion

Smallholder farmers of the Kaziranga region pursue a variety of livelihood strategies in addition to mixed farming enterprises. Their farming strategy is dependent on the seasonal inundation from the Brahmaputra, but their homes and livelihoods are also at risk from the same flood. A variety of animals are kept, according to village conditions and traditions, allowing efficient use of marginal farm and village resources. The desirable traits selected in livestock by an Assamese farmer, would, barring some climatic variations, be familiar to a farmer living in Scotland. Characteristics associated with productivity and survivability were given primacy over aesthetic considerations.

Meeting groups and interviewees tended to discuss disease syndromes affecting their livestock, rather than issues of grazing and fodder. The effects of the seasonal flood were often discussed, and were considered to be an immutable feature of life that had to be lived with, rather than a problem to be overcome. Crop raiding, and other conflict with wild animals, was also frequently discussed, but participants chose not to include these issues when asked ". . . to discuss the problems that limit how much your animals contribute to your household" Similar to a study conducted in Himachal Pradesh [26] participants prioritised the discussion of animal health problems and infectious diseases, at least in part because they did not normally have access to a veterinary clinician to discuss these matters with. In the current study it is likely that participants were chose to focus on discussion of livestock disease syndromes as a consequence of the facilitators introducing team members as veterinary doctors.

Village meetings were well attended by women and men, and individuals of varying wealth and social status within the village, allowing for a reasonable representation of differing opinions within the village. Not all meeting participants presented themselves equally, however facilitation by experienced local community workers from TCF created the space for all participants to be heard, and sought to minimise bias resulting from intra-community factors of social status, relative affluence, and gender. The team included a female veterinary clinician, however it would have been preferable if female community workers had also been available, as this would assist in enabling women's voices to be fully heard.

Farmers contributed eagerly to open discussions on animal health, sharing their experiences and concerns. In common with the experiences of Ayele et al [28], Wright and Thrusfield [26], and Catley et al [29] proportional piling exercises were undertaken enthusiastically by participants, with good-natured debate as to the relative importance of different challenges. The discussion of these results and re-piling or re-ranking that followed resulted in lively discourse between neighbours before ultimately arriving at a consensus.

During the pilot meetings for this project farmers were unwilling to undertake pairwise comparison exercises, preferring to rank the conditions in order of importance; in the words of one male farmer: *"Look, we've put the problems in order, please stop picking them up two at a time. We would like to spend time discussing the conditions more."* Likewise, during the pilot meetings using a matrix scoring grid to match the importance of clinical signs to conditions also proved impractical as participants tended to add more and more symptoms. Instead participants preferred to describe conditions in their own words, and these descriptions were then coded for analysis. This differs from the experiences of Bardsley and Thrusfield [30] and Catley et al [31], who describe farmers in other regions engaging readily with these activities.

Farmers diverged from the researcher's plans during several meetings, either by re-ranking the conditions or re-piling the stones such that the two activities no longer matched. While this complicates the analysis of the information, to be truly participative meetings must follow the path preferred by the participants, not follow unerringly the course that has been pre-set by the organisers. This is key to the participants taking ownership of the programme and maximising their engagement with subsequent interventions.

Farmers were very keen to be interviewed at their homes, and the team were unable to meet with every household that wished to take part. Interviewing farmers at their homes allowed them to speak freely without worrying about the judgement of their peers, however each interview is inherently the opinions of a single individual or small group. Interviews with a pair of respondents are particularly valuable as the participants' moderate each other's views and discuss and debate between themselves their recollections of past events, the significance of current issues, and future hopes fears and plans. Respondents found the experience of being interviewed extremely novel, and once they became accustomed to it, relished the opportunity to discuss farming and animal health with someone from another place. Respondents were grateful for the opportunity to represent their views through participation, indicating that they normally felt excluded from consultative or decision-making processes. One female farmer stated: *"It is good that you are here. Normally no one bothers to listen to us, to ask us what we think is important."* An aside from another female farmer to her husband during an interview: *"They are actually listening to us, and writing it down!"* Participants were very keen to learn more about animal health and the treatment of disease, making themselves both more self- sufficient and more productive. In the words of one male farmer: *"I would like to learn from a Doctor or compounder* (paravet*) how to treat my own animals. . . Practical teaching. Meetings would be very useful. I could learn something from leaflets also. . . No one ever teaches us. . . Because it is for me I will have to manage time, and I will manage."* Participants displayed a strong sense of responsibility and expressed firm desire to increase their animal heath skills. Another male farmer positively asserted about animal health education: *"I can do this for my own cattle and my community."*

Participants will often ascribe a higher priority to events which occurred recently than those that occurred some time ago, as the impact of these events is fresher in their memories. Disease outbreaks affecting a large proportion of animals over a short period of time, particularly those with a high mortality rate or dramatic clinical signs, can often have a more powerful impact on the perceptions of farmers than those conditions of an endemic, ubiquitous, or management aetiology which continually erode animal health and productivity, or cause sporadic deaths. Both FMD and goat pox were assigned a high priority by respondents. Recent outbreaks of both diseases had occurred before the time of data collection. This tendency is similar to the observations made by Bellet et al [32]. FMD outbreaks are common in the region, along with reactive vaccination programmes focused on the villages close to the KNP. Goat pox was however a novel disease in the villages (despite being classes as endemic in the region), and this condition raised a high level of fear and uncertainty among farmers [14]. As a result of these investigations the presence of goat pox was reported to the local authorities and a vaccination programme initiated.

Participants were adept at recognising changes in their animals and were most observant of patterns of clinical signs, in keeping with the observations of Ayele et al [28]; Catley et al [33]; and Bardsley and Thrusfield [30]. Conditions with clear and consistent clinical signs will tend to be identified more readily than those with variable, complex or subtle symptoms, and thus might be construed to be more common or important, and therefore be assigned a higher priority by participants. Wright and Thrusfield [26] caution against over interpretation of vague clinical signs, and Ayele et al [28] acknowledge that one must always be aware of the limitations of the diagnostic ability of livestock keepers. Similarly, certain animal species may be

considered 'more important' for reasons not directly relating to their economic activity or value, so diseases affecting these species may be considered more important than a disease of greater overall impact affecting another species. For example, meeting group D initially excluded Ranikhet from the piling exercise because it is a disease of poultry, but later placed it high in the ranking exercise. Male respondents in particular tended to focus on conditions affecting cattle, oxen and goats rather than those suffered by chickens, ducks or pigeons. Female farmers were much more likely to discuss problems affecting poultry, calves and goat kids than their male counterparts; women are frequently the main carers of poultry, young animals and small ruminants. Grace et al [34] also comment on the differing information gained from talking to female livestock keepers compared with male.

## Conditions of importance

Villagers generally describe animal health conditions accurately and with considerable detail, and these descriptions are generally repeated in very similar terms between individuals and villages, this is in keeping with the findings of Bardsley and Thrusfield (30) and Catley et al [31]. Some conditions: FMD (*Chaboka / Kurra phatta*); Newcastle disease (*Ranikhet*); Goat pox (*Sagolay bohonta*); sarcoptic mange (*Papora dhora*); were not only eloquently and consistently described by participants, clinical cases were also examined by the team, leading to a high level of confidence in the diagnosis. Other diseases such as black quarter (*Moh-bis-oni*); listerial meningoencephalitis (*Murigoni*); and anthrax (*anthrax*), are very well described by numerous participants, making these the likely diagnoses for these local terms.

Some of the disease terms used by participants referred to a clinical sign which encompassed a group of conditions. *Hagoni* is a general term for diarrhoea or dysentery. The presentation of most concern to animal keepers is a rapidly fatal disease of goats, particularly the young, grazing new grass after the seasonal flood. The description closely resembles clostridial enterotoxaemia, however the possibility of a parasitic mass emergence should not be discounted. Diarrhoea of young calves, such as rotavirus, is also captured by the term *hagoni*, as is diarrhoea of dietary origin. Parasitic gastro-enteritis is also included in this group of conditions, although unless worms are observed in the dung, the cause is not recognised by the farmers. There are numerous other potential infectious and non- infectious causes of *hagoni*.

*Gol phulla* is used to describe a swollen neck, however this appears to encompass two conditions, one resulting in rapid death and another causing a progressive condition loss. The terms *Bhekulia Dhoka dingra*, *Dingi phulla* are also used to describe conditions involving a swollen neck, however there was variability between participants about which term was used, and which disease it described. The most likely interpretation of these terms is that the rapidly fatal form represents haemorrhagic septicaemia and the wasting disease is chronic fasciolosis, though particularly for the condition loss it should be appreciated that there are other conditions (particularly hepatic or cardiac in origin) that could result in the syndrome described.

**Conditions of unexpectedly low importance.**   Some conditions are notable for their surprising absence from farmers' discussion, or low level of concern expressed. There was no discussion of anaemia; respiratory disease was not discussed, other than as a per-acute condition; mastitis of cattle was not raised by participants, only agalactic mastitis in goats.

Ticks (*sikora*) were discussed in two meetings and three interviews. This probably indicates a low level of parasitism by ticks, as experience elsewhere in India found that owners readily observe ticks, and when present assign a priority to them [35]. It is possible that the local conditions of seasonal inundation reduce tick populations. It is further possible that the low level of observed ticks results in a lower level of blood parasites than seen in some areas, explaining the lack of discussion of anaemia among respondents.

Worms (*pelu*), in terms of gastro-intestinal nematodes and cestodes, were only discussed by two meetings and one interviewee. As observation of worms in faeces is the main method of identification it is likely that intestinal parasitism is under-recognised by respondents. Given the marked clinical signs associated with *Toxocara vitulorum* in calves, it could be that this parasite is less prevalent among these animals compared with a previous study in Madhya Pradesh [35]. Surprisingly, the presence of haemonchosis in young animals did not prompt further discussion of anaemia, which has been noted commonly among goats in the locality [36].

Only one meeting group discussed poor weight gain in goats or calves, and although fertility issues were often discussed, no group raised them as a priority. This could indicate low expectations of the productive capabilities of animals as a result of farmers having long term experiences of suboptimal productivity, effectively normalising this situation. A concurrent study in the region found that the majority of goats were underweight [36], so it is possible that not identifying poor growth rates as a concern represents a lack of recognition of the production potential of these animals, or a resignation to the availability of nutritional resources for livestock as a limiting factor. These attitudes could lead farmers to be content with inefficient of production when they could aspire to more.

## Comments on socio-economic situation and change

The increasing educational attainment of village children, combined with an existing culture of seeking additional employment out with the family farm, makes it likely that in the future more villagers will pursue their main employment outside the village, potentially leading to a decrease in the proportion of the village population identifying themselves as farmers, ultimately following the trend of increasing urbanisation of the population. This could result in a decrease in the number of agricultural units as the remaining farmers consolidate agricultural land. Pre-existing systems of land sharing or rental, such as *adhi*, could play a role in this process.

An agricultural system which is both dependent on, and at risk from, a seasonal flood, is extremely vulnerable to the effects of the climate crisis [13]. In addition to this, the loss of natural habitats, and changing patterns of rainfall, is resulting in increasing human- wildlife conflict in the region. In response, the farmers of the future in this region will have to be more adaptable in order to run productive units, however with the limited resources currently available this could be challenging to achieve. Income, access to agricultural extension, and access to credit have been found to be key factors in the ability of farmers to adapt their farming techniques to mitigate the effects of the climate crisis on their farms. Farm diversification, including increasing livestock numbers, is one strategy to mitigate for the effects of increased climate unpredictability as a result of the climate crisis [8].

## Comments on the interpretation of findings

The triangulation of data through consideration of information from a variety of sources allows for interpretation of data to maximise the reliability of findings and their relevance to the local human and animal population and the regional situation.

Formulating a list of farmers' animal health concerns is not the same as assessing the true prevalence of disease or financial cost through classical epidemiological or economic techniques. However, these numerical measures do not properly capture the impact of animal diseases on smallholder households, highlighting the value of the techniques outlined in this manuscript. Key to engaging stakeholders and giving them ownership of any programme is addressing their priorities, determined through participatory discussion and investigation. This enables farmers to consider animal health as a community issue, working with researchers and clinicians to identify problems and appropriate local solutions [8, 33].

## Suggested mitigation strategies

The data presented in this study indicate opportunities for impactful interventions such as planned animal management, immunisation programmes, and selective use of anti-parasitic drugs. These could help to improve animal health, profitability, and sustainable land use. Conditions considered of greatest importance by the participants include four specific, vaccinatable, infectious disease plagues: FMD, haemorrhagic septicaemia, Newcastle disease, and goat pox; three parasitic diseases: liver fluke, sarcoptic mange and parasitic gastro- enteritis; and three primary clinical signs: diarrhoea, bloat, and neck or throat swelling. Currently in the region vaccination is usually carried out for FMD only, usually on an ad-hoc basis in response to outbreaks. Planned programmes of vaccination for foot and mouth disease, haemorrhagic septicaemia, goat pox and Newcastle disease would have the potential to yield great benefits, while biosecurity education could help to limit the introduction and spread of these diseases. Education on care of neonatal ruminants, and particularly adequate colostrum feeding, could help reduce the incidence of diarrhoea and other causes of mortality and ill thrift in young animals [15]. Grazing management education could assist in the control of flukes, gastro-intestinal nematodes, and some causes of diarrhoea and bloat, as well as leading to more efficient use of available grazing resources, and this can be combined with the targeted use of anti- parasitic medications under the guidance of veterinary clinicians or paraprofessionals. The success of these interventions depends on proper engagement of the community; and initiatives must be adapted to local community needs and conditions. Participants indicated a preference for face to face practical training, but also acknowledged the benefits of printed education materials in the local language. Linking health interventions, such as immunisation, to impactful farmer education, including the equal engagement of women farmers increases the potential for wide-ranging gains by improving animal health and productivity

## Supporting information

**S1 File. *Aide-mémoire* for in-depth interviews.**
(DOCX)

**S2 File. Disease descriptors and their frequency of use.**
(DOCX)

**S3 File. Details of village meetings, attendance, animal species kept, disease descriptors, breakdown of proportional piling exercises, and interview notes.**
(DOCX)

**S1 Table. Household livelihood steams of interviewees.**
(DOCX)

## Acknowledgments

The authors would like to thank the people of the villages surrounding the Kaziranga National Park, without whose assistance this work would not have been possible. We would also like to thank our colleagues at The Corbett Foundation and the Royal (Dick) School of Veterinary studies for their help and understanding with the undertaking of this project.

## Author Contributions

**Conceptualization:** Andy Hopker, Naveen Pandey, Sophie Hopker, Rebecca Marsland, Neil Sargison.

**Data curation:** Andy Hopker.

**Formal analysis:** Andy Hopker, Rebecca Marsland.

**Funding acquisition:** Neil Sargison.

**Investigation:** Andy Hopker, Naveen Pandey, Sophie Hopker, Dibyajyoti Saikia, Jadumoni Goswami, Roopam Saikia, Sumanta Kundu.

**Methodology:** Andy Hopker, Sophie Hopker, Rebecca Marsland, Michael Thrusfield, Neil Sargison.

**Project administration:** Naveen Pandey, Neil Sargison.

**Resources:** Naveen Pandey, Dibyajyoti Saikia, Roopam Saikia.

**Supervision:** Rebecca Marsland, Neil Sargison.

**Validation:** Michael Thrusfield.

**Writing – original draft:** Andy Hopker.

**Writing – review & editing:** Naveen Pandey, Sophie Hopker, Dibyajyoti Saikia, Jadumoni Goswami, Rebecca Marsland, Michael Thrusfield, Neil Sargison.

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
