## [Decision Letter · Decision Letter 0]

1 Jun 2020

PONE-D-20-12276

Animal health perceptions and challenges among smallholder farmers around Kaziranga National Park, Assam, India: a study using participatory epidemiological techniques.

PLOS ONE

Dear Dr. Hopker

Thank you for submitting your manuscript to PLOS ONE. After careful consideration, we feel that it has merit but does not fully meet PLOS ONE’s publication criteria as it currently stands. Therefore, we invite you to submit a revised version of the manuscript that addresses the points raised during the review process.

Many thanks for submitting your manuscript to PLOS One

There was significant disagreement between the first two reviewers, so I invited a third reviewer to act as a decider.

They have made a number of recommendations for modification prior to acceptance

If you could write a detailed response to reviewers, that will aid to expedite review when you resubmit

I wish you the best of luck with your revisions

Hope you are keeping safe and well in these difficult times

Thanks

Simon

We look forward to receiving your revised manuscript.

Kind regards,

Simon Clegg, PhD

Academic Editor

PLOS ONE

Journal Requirements:

3. Thank you for stating the following at the beginning of your manuscript:

'This work was funded by the Royal (Dick) School of Veterinary Studies, University of

Edinburgh, as part of the Indian Veterinary Education Project.'

We note that you have provided funding information that is not currently declared in your Funding Statement. However, funding information should not appear in any areas of your manuscript. We will only publish funding information present in the Funding Statement section of the online submission form.

'The author(s) received no specific funding for this work'

Reviewers' comments:

Reviewer's Responses to Questions

**Comments to the Author**

1. Is the manuscript technically sound, and do the data support the conclusions?

Reviewer #1: Partly

Reviewer #2: Partly

Reviewer #3: Partly

2. Has the statistical analysis been performed appropriately and rigorously? 

Reviewer #1: N/A

Reviewer #2: N/A

Reviewer #3: No

3. Have the authors made all data underlying the findings in their manuscript fully available?

Reviewer #1: Yes

Reviewer #2: Yes

Reviewer #3: No

4. Is the manuscript presented in an intelligible fashion and written in standard English?

Reviewer #1: Yes

Reviewer #2: Yes

Reviewer #3: Yes

5. Review Comments to the Author

Reviewer #1: This is an interesting study with a good and relevant design that elucidates important aspects of livestock production in LMICs.

One general question I have is whether the study participants had been exposed to any previous information campaigns, animal health programmes or official disease control interventions? A sentence about this would be useful to add in the Introduction section.

It is customary to phrase figure captions and table headings so that they stand alone, without having to refer to the text to understand the setting and methods. This is not the case in this manuscript, but I leave it to the editor to decide whether any changes are necessary.

Specific comments are listed below.

Abstract

Line 27: For clarity, I suggest changing flood to flooding

Introduction

Line 47: I suggest adding approximately or similar in front of the estimated number of holdings

Line 50: sentence begins with figure, please change to Eighty-six percent, or insert a word first.

Line 70: Please clarify what is indicated by “consideration” of livestock husbandry, as this is not clear to me

Line 75: Please replace “of” by “for” (used for fodder production)

Line 76: Please delete s in limits (plural)

Line 98: Please replace Uniting by United (Nations)

Line 99: I suggest removing “activities” to simply read “productivity of farming,”

Lines 105-106: What is the broader aim referred to here?

Lines 108-111: Please remove the last sentence listing methods, this belongs to Materials and Methods

Materials and Methods

Line 128: Do you mean “…the highway to the south” or”… the highland in the south”?

Lines 129-130: Please describe briefly how you came to know of the villages before selection, was there a list/register? Did you know beforehand of the land types, farming styles and population that you mention and, if so, how?

Line 145-146: Were the participants informed that they could withdraw from the study if they wished to?

Line 162: I suggest writing “..as previously described (17, 18, 19)” instead of “described by (17, 18, 19)”

Line 163: Either add a plural s to “group” or change “were” to “was”

Line 169: the submission guidelines only state English, not necessarily AmE or BrE, but most of the manuscript appears to be written in BrE, except the word draft (although spelled draught on line 368). I suggest checking the text for consistency as regards type of English.

Line 186-187: the reference is not cited correctly

Line 214: For better understanding, I suggest adding a sentence on the interview training/experience of the two people conducting the interviews, and how they presented themselves to the interviewees (positioning, introduction about who was in charge/conducting the study etc).

Line 235: I suggest changing to “…and to provide feedback of the study findings…”

Results

I suggest organising the results section in the same order as the M&M section, as this makes it easier for the reader to grasp how the results are linked to the methods.

Lines 258-260: Please swap the order of the last two sentences to finish the paragraph with the statement about interesting conversations and perspectives, instead of the age of the participants.

Line 267: Please delete “the” before “in the tourist industry”

Line 270: I find the phrasing “access additional income from the home” a little confusing, do you mean that they can do the work (and sales) in their house or that they can get the income from resources in the household?

Line 293: Please delete “as” after “including”

Line 343: I suggest adding a hyphen between “human” and “wildlife” (human-wildlife instead of human wildlife)

Line 386: Is it possible to go back to the original language and suss out what is indicated by “look” in the cited statement? It would be interesting to understand if the person is talking about the prettiness of the animal, the physical state of the animal or the brightness of the eyes of the animal. From the subsequent text I would guess they talk about if the animal is pretty or ugly but this is also a concept that is hard to understand unless you know about their views on that. (fat/skinny and woolly/shiny can also be part of the concept of beauty…)

In addition, could you specify if they mentioned signs of disease (or rather, lack of them) when talking about a good animal?

Line 392: Pleas delete “-ly” to read “considered positive” (unless I have misunderstood the sentence

Lines 423-426: If village D did not want to include Ranikhet (please explain here that this is probably ND), how did it come up in the first place? I had the impression that the participants suggested the key challenges themselves, without prompting? If this was not the case, please clarify in M&M

Line 436: please delete “that”

Line 438: how can this be table 2, where is table 1?

Line 444: I suggest moving this section to follow directly after the section Animal health challenges, to improve understanding.

Please format the table so that it is obvious what are column headings

Discussion

Line 467: Please change climactic to climatic

Line 468: Don’t you mean aesthetic instead of ascetic?

Lines 469-470: Didn’t you lead them into discussing disease syndromes? Was this really the case, unguided? If the group discussions preceded the interviews and they recognised you from the in-depth discussions about the importance of different diseases, could this have affected the way the responded?

Line 474: As they did not normally have access to a veterinarian, how did you introduce yourselves in the discussions, could the mentioning of a veterinary degree in one of the interviewers have affected the line of conversation? I think this is what you are hinting at and I suggest making it more clear.

Line 478: Please specify the gender of the CKWs. I suggest a brief reflection on whether this could have mattered for the discussions.

Line 515: The positive quote indicates that the participants may have expected your listening to them to lead to more than a scientific paper, was this discussed or acted upon?

Lines 522-523: Were any official control measures prompted by the outbreaks and, if so, could this have affected the responses?

Line 539: I suggest clarifying that female farmers are more likely to be the ones caring for these animal species and hence more likely to discuss them.

Line 540: Please remove s from comments (Grace et al – plural)

Line 546: couldn’t Ranikhet also be avian influenza?

Line 550: Please remove “and” before “making them…”

Line 585: I suggest using the past tense (did instead of does), to avoid confusion, and to specify if you refer to the current study or a previous one (I assume that the reference is a previous study but this could be written more clearly)

Line 587: From other LMIC studies, it appears that fertility issues are not recognised as “diseases” or “problems” if they are common. Perhaps a comment on the different views on problems that occur out of the ordinary and are regarded as novel as compared to things that are more or less expected would be useful?

Lines 598-599: I suggest inserting “themselves” between “identifying” and “as farmers” and “the” between “urbanisation of” and “population”.

Line 609: This is a very bold statement, are you sure the information is an accurate representation of the participants’ perceptions? I would prefer a more modest phrasing, as the perceptions of the participants were presented to you in a certain situation, and interpreted by you.

Line 621: Again, are you sure the data highlight opportunities for interventions, did the participants say they wanted this? Or do you mean that the data indicate that there is a potential for improvement and that interventions such as those you suggest could, if implemented, improve animal health etc.? I may have missed this but I haven’t seen any direct results on suggested interventions and how implementation of different interventions would be easy to achieve.

References

Not all references appear to entirely follow the journal instructions, please check and revise where relevant.

Reviewer #2: The article describes a participatory rural appraisal conducted in Kaziranga region, Assam state, India, to identify the main animal health issues affecting smallholder livestock farmers and explore their impact.

While the study itself is interesting, the manuscript, as it stands now, is too descriptive to qualify as a scientific article. Based on the methods and themes raised by the authors in the interviews (lines 161-170) more results and discussion are expected on the impact of animal health issues, and natural hazards like flood, on farmers’ wellbeing, distinguished by their different effects, e.g. on nutrition, income, disturbance of farming activities and stress, and farmers’ attitude towards the risks. No such results are displayed and the authors do not explain the added value of their method compared to more classical impact assessments (stated in lines 614-615). Instead, they only list the health challenges recognized as “important” by farmers, but without explaining what this importance is based on. Similarly, the authors do not make any real discussion on policy recommendation. The section “mitigation strategies” (lines 621-626) is very short and does not provide any new ideas for promoting interventions aims at improving animal health in smallholder systems: what are the most appropriate mitigation strategies? How mitigation strategies could be best implemented? What communication strategies would be effective?

Additionally, some methodological points deserve more explanation :

- The authors should describe the purpose of this ranking exercise: is it in order to confirm the outcome of the proportional pilling exercise?

- Proportional piling scores on the importance of animal health issues are presented as a critical result of the study. But the authors do not explain on which criteria the score is based: severity of the disease, frequency, production loss, disturbance of farm activities? How are these different effects accounted for and combined to compute the final score? This deserves more explanation.

- In-depth individual interviews: were the three villages part of the ten villages where the authors conducted the village group meetings? What is their reference (A - J?)? How were they selected? Why only three villages? How were the households selected for in-depth interviews? What was the method for conducting interviews (semi-structured with the help of a checklist? Unstructured?)? If a checklist was used, it should be available in an appendix.

- In the list of identified diseases and condition names with likely diagnoses, which ones were confirmed through clinical examination?

- Regarding gender differences: the authors should indicate whether females and males were interviewed separately and discuss the interest of conducting separate interviews.

In the results section: for each piece of information given in this manuscript (e.g. “The size and physique of an ox were considered particularly important, especially the musculature of limbs”), it would be good to indicate how many collective and individual interviews the information was extracted from. The authors should provide precise figures whenever possible rather than vague terms like "often".

Other minor comments:

- Lines 469-474: do you have any figure showing the importance of animal health problems for farmers? If yes, include it in the result. Otherwise remove this sentence because it is not supported by data.

- On lines 477-487, the authors should avoid terms with connota

Reviewer #3: This study is interesting and contributes to understanding the livelihoods, environmental challenges and livestock diseases of the small owners who live in the vicinity of a national park. Scientifically, it is a subject that deserves to be studied, however, the manuscript needs a lot of work to achieve a suitable version to be published in Plos-one. It is a difficult manuscript to evaluate, although the English language seems correct, the organization of the manuscript is poor and, in some cases, chaotic. Although the data is descriptive and qualitative in nature, the authors should explain how they analyzed it based on the objectives of the study, and not on the methodological tool. It is not clear how the information obtained from the various techniques used was combined. In the figures the authors have not translated into English the diseases described by the respondents. The bibliography is scarce and does not cover the basic concepts on which the study is based. Authors should make an effort to find adequate bibliography.

Abstract

The summary is discursive and does not correctly inform about the aim of the study, the methodology used, particulars of the participants and other characteristics of the study that allow the reader to know the real value of the study. Authors should develop a summary that is autonomous enough from the manuscript to understand the nature and value of the study.

Introduction

The authors dedicate the first four paragraphs of the introduction to the particular problems of the study area. This start gives a limited and local character to the study. Only the fifth paragraph reviews some generic farm animal health topics. The authors should rewrite this section starting with the problems that small producers generally face: 1) climate change or climatic adversities; 2) the socio-environmental conflicts and livelihoods of the populations that live near or are immersed in the national parks (i.e. Estevez-Moreno et al 2019 small ruminant research); 3) animal health and production under the conditions of smallholders, and 4) the importance of smallholder perceptions for future improvements in public policies. Later, they can explain the particularities of the study area, along with the justification, hypothesis and objectives of the study. This order will improve the reader's understanding of the conceptual framework of the study and the applicability of the study to the understanding of rural life not only in India but also in other similar agro-ecosystems in the world.

M&M

L116-133 The study area is described descriptively and in a colloquial style. The authors do not give accurate information about the geographical conditions, the climatic categories (i.e. Köpen classification), topographic and other information that allows the reader to understand the study environment. All three maps have poor resolution and are not very helpful in understanding the study environment.

L133 Authors should develop a paragraph explaining the study design that includes all stages and the sequence for all participatory techniques. There is no sociodemographic information that allows us to understand the population studied.

141-143 I do not understand the usefulness of individually identifying each member of the team.

L244-260 These are not results, this paragraph must be in M&M.

I have decided not to comment on the results and the discussion. I encourage the authors to make the changes to the introduction and M&M so that in a second revision they can fully understand the manuscript.

6. PLOS authors have the option to publish the peer review history of their article (what does this mean?). If published, this will include your full peer review and any attached files.

Reviewer #1: No

Reviewer #2: No

Reviewer #3: No

---

## [Author Response · Author response to Decision Letter 0]

20 Jul 2020

Dear Dr Simon Clegg and Reviewers

We hope this e-mail finds you all well, and that you have not personally been too adversely affected by the global pandemic.

Thank you all for your constructive comments on our manuscript. We very much appreciate the time taken by the reviewers to provide such detailed feedback. We have addressed each point raised, and I hope that our alterations meet with your approval. 

We have detailed the changes made below, with our response and alterations to each point raised by the reviewers.

We feel that your input has strengthened the manuscript, I hope that you agree, and that you now find the work suitable for publication in PLoS ONE.

Please contact us should further alterations be required.

Yours faithfully

Andy Hopker, Naveen Pandey, Sophie Hopker, Rebecca Marsland and Neil Sargison

Alteration have been made to formatting and file naming where appropriate. I hope that these changes meet with your satisfaction.

This has been done, and the aide-mémoire is now included in supplementary materials

3. Thank you for stating the following at the beginning of your manuscript:

'This work was funded by the Royal (Dick) School of Veterinary Studies, University of

Edinburgh, as part of the Indian Veterinary Education Project.'

We note that you have provided funding information that is not currently declared in your Funding Statement. However, funding information should not appear in any areas of your manuscript. We will only publish funding information present in the Funding Statement section of the online submission form.

'The author(s) received no specific funding for this work'

Altered as requested

Done

5. Review Comments to the Author

Reviewer #1: This is an interesting study with a good and relevant design that elucidates important aspects of livestock production in LMICs.

One general question I have is whether the study participants had been exposed to any previous information campaigns, animal health programmes or official disease control interventions? A sentence about this would be useful to add in the Introduction section.

The following statements have been added to the introduction.

“Access to agricultural extension education is variable in the region, and extension activities tend to be directed towards cultivation rather than livestock (Goswami, 2015; Begham and Manhata, 2017). Livestock immunisation programmes are largely reactive in nature, vaccinating animals in the face of an outbreak.”

It is customary to phrase figure captions and table headings so that they stand alone, without having to refer to the text to understand the setting and methods. This is not the case in this manuscript, but I leave it to the editor to decide whether any changes are necessary.

Captions have been altered to improve clarity

Specific comments are listed below.

Abstract

Line 27: For clarity, I suggest changing flood to flooding.

 Throughout the manuscript “flood” has been variously altered to “inundation”, “seasonal flood” or “flooding” other than where quoted, as inhabitants simply say “the flood”

Introduction

Line 47: I suggest adding approximately or similar in front of the estimated number of holdings Done

Line 50: sentence begins with figure, please change to Eighty-six percent, or insert a word first. I have rephrased this sentence

Line 70: Please clarify what is indicated by “consideration” of livestock husbandry, as this is not clear to me I have removed this sentence.

Line 75: Please replace “of” by “for” (used for fodder production) Done.

Line 76: Please delete s in limits (plural) Done

Line 98: Please replace Uniting by United (Nations) Done

Line 99: I suggest removing “activities” to simply read “productivity of farming,” Done

Lines 105-106: What is the broader aim referred to here? Now reads: “…broader aim of achieving sustainable improvements to animal welfare and productivity”

Lines 108-111: Please remove the last sentence listing methods, this belongs to Materials and Methods Done

Materials and Methods

Line 128: Do you mean “…the highway to the south” or”… the highland in the south”? altered to: “..access the highground in the south”

Lines 129-130: Please describe briefly how you came to know of the villages before selection, was there a list/register? Did you know beforehand of the land types, farming styles and population that you mention and, if so, how?

The following statement has been added to the section:

“This convenience sample utilised pre-existing contacts with village inhabitants to organise village meetings.”

Line 145-146: Were the participants informed that they could withdraw from the study if they wished to? Yes. Thank you for reminding me of this, the following statement has been added:

“Participants were informed that they could withdraw from the project if they wished, before asking if they wished to continue.”

Line 162: I suggest writing “..as previously described (17, 18, 19)” instead of “described by (17, 18, 19)” Done

Line 163: Either add a plural s to “group” or change “were” to “was” Done

Line 169: the submission guidelines only state English, not necessarily AmE or BrE, but most of the manuscript appears to be written in BrE, except the word draft (although spelled draught on line 368). I suggest checking the text for consistency as regards type of English. Corrected

Line 186-187: the reference is not cited correctly Corrected

Line 214: For better understanding, I suggest adding a sentence on the interview training/experience of the two people conducting the interviews, and how they presented themselves to the interviewees (positioning, introduction about who was in charge/conducting the study etc). The following descriptions have been inserted: 

“Each interview was arranged in advance with the help of community members. The researchers arrived at each household and waited to be invited in. All participants (farmers and researchers) typically sat in a loose circle at the same level. The community worker opened each interview with introductions, followed by description of the purpose and methods of the study, answered questions about the study, and offered the opportunity to withdraw, prior to taking and recording informed verbal consent from the farmer participants.” 

And:

“The veterinary clinician posed each opening question, and then both researchers engaged in the ensuing discussion in a conversational manner.”

Line 235: I suggest changing to “…and to provide feedback of the study findings…”

The following insertion has been made:

“Interview participants attended village participatory meeting A. Meetings I and J, though in the same cluster of villages, were attended by different farmers. A further three meetings were held for the interviewees, to report the findings of the interviews, provide relevant animal health education and advice on community animal health planning, in keeping with the aims of the study. Further relevant animal health education is planned for the other villages in the region, based on the findings of the study.”

Results

I suggest organising the results section in the same order as the M&M section, as this makes it easier for the reader to grasp how the results are linked to the methods.

The results have been organised to describe various salient points about the human and animal population, farming strategies and environmental challenges first in order to provide the reader with relevant background, before embarking on description of animal health challenges. At each stage all relevant information acquired during the study is draw in to the discussion of that point

Lines 258-260: Please swap the order of the last two sentences to finish the paragraph with the statement about interesting conversations and perspectives, instead of the age of the participants. Done

Line 267: Please delete “the” before “in the tourist industry” Done

Line 270: I find the phrasing “access additional income from the home” a little confusing, do you mean that they can do the work (and sales) in their house or that they can get the income from resources in the household? Deleted “from the home”

Line 293: Please delete “as” after “including” done

Line 343: I suggest adding a hyphen between “human” and “wildlife” (human-wildlife instead of human wildlife) done

Line 386: Is it possible to go back to the original language and suss out what is indicated by “look” in the cited statement? It would be interesting to understand if the person is talking about the prettiness of the animal, the physical state of the animal or the brightness of the eyes of the animal. From the subsequent text I would guess they talk about if the animal is pretty or ugly but this is also a concept that is hard to understand unless you know about their views on that. (fat/skinny and woolly/shiny can also be part of the concept of beauty…)

In addition, could you specify if they mentioned signs of disease (or rather, lack of them) when talking about a good animal?

An interesting and useful point. “Look” means prettiness in this context, I have added this to the quote in brackets. Incidentally in this region, fatness and shininess would be considered both ‘good’ and ‘attractive’ features. Wooliness was not a state I encountered- presumably to hot and humid for such animals to thrive.

Feature of health (therefore the absence of feature of ill health) were frequently discussed as good. 

“the hoof should be pointy, tight shape, upright, with a small gap (between the toes)…”

I would interpret the above quote as quite informed- as inspection of the foot is being used as an indicator of age and to assess for freedom from: current FMD infection; lasting effects of FMD; interdigital necrobacilosis; white line disease; upper limb lameness; serious ill health events (stress lines); specific nutrional deficiencies.

I have moved additional testimonials, with a brief explanations, from the supplementary materials to the main body of the manuscript:

“Colour, health, physique. Ears erect, eyes. You know if an animal is healthy by looking, legs are muscular, the hoof should be pointy, tight shape, upright, with a small gap (between the toes)… Colour- red and black mix, no white. White is not so good, not suitable for this home… Cows should have a strong rump, if the cow and calf are together, then look at the health of the calf… To buy animals (my father in laws’) sons all go together. Sometimes an experienced man from the village goes with them if available.”

Interviewee 18, female 22

Five male interviewees said that they would have an experienced or knowledgeable man from the village accompany them when they went to purchase a cow or oxen. It seemed that this service would be freely offered by a friend or neighbour, however the man receiving advice might, for instance, provide lunch for his neighbour.

“The animal’s legs, walking style, health and body size are all important, but I don’t have that knowledge. I only look at its legs and its walking style and I guess if it is a good animal. I take an experienced man from the village with me when I go to buy cattle, and I listen to what he says.”

Interviewee 11, male, 58

 Two female interviewees described their husbands as knowledgeable men whose help might be sought in the selection of new livestock. Two respondents discussed the importance of questioning the vendor about the milk production of the cow or the cow’s mother, while another said that it was important to see a cow’s previous calves if possible. 

Other considerations included hair colour, though participants were careful to point out that this was a secondary factor. The preferred (or avoided colour) varied between houses (or households?) within the village.

“Colour (of the animal) should suit the house. Black and red are right for this house. Colour is not the main consideration though, other things must come first.”

Interviewee 4, male, 27.

The animal’s eyes, facial structure, hair quality and horn shape (or absence) were also considered by some respondents, but only as secondary factors. A long tail was also considered an advantage to keep off flies. Six interviewees mentioned aesthetic considerations such as colour or facial features, however four of these interviewees stressed that these were of secondary importance only. 

Line 392: Pleas delete “-ly” to read “considered positive” (unless I have misunderstood the sentence

I have altered the sentence as follows:

“Increased height was a positive factor in draught oxen”

Lines 423-426: If village D did not want to include Ranikhet (please explain here that this is probably ND), how did it come up in the first place? I had the impression that the participants suggested the key challenges themselves, without prompting? If this was not the case, please clarify in M&M

This has been clarified by extending this section as follows:

“Village D discussed Ranikhet, a disease of chicken and ducks, at length during the open discussion, but did not want to include Ranikhet in the proportional piling exercise as they identified themselves as cattle and goat farmers. However, when asked to rank the conditions considering the financial impact, they placed sagolay bohonta in position 1 and Ranikhet in position 2. Conditions were re-ranked, but piling results were not altered by the group.”

Line 436: please delete “that” Done

Line 438: how can this be table 2, where is table 1?

Corrected. Sorry for the mistake, a bonfire of the tables occurred during the writing process!

Line 444: I suggest moving this section to follow directly after the section Animal health challenges, to improve understanding.

Please format the table so that it is obvious what are column headings

I have increased the weight of some lines and increased the font size of the headings (by 1 point only). I hope that this makes the table clearer.

Discussion

Line 467: Please change climactic to climatic done. Thank you.

Line 468: Don’t you mean aesthetic instead of ascetic? Done. Thank you.

Lines 469-470: Didn’t you lead them into discussing disease syndromes? Was this really the case, unguided? 

The opening statement to the group was “We have all come here today to discuss the problems limit how much your animals contribute to your household” however, immediately before this I was introduced to the group as “A veterinary doctor from Scotland who has worked in rural India for some time.” Thus while I didn’t directly lead them into discussing disease syndromes, I caused this to occur. I have altered both the M&Ms and the discussion accordingly to better capture this.

If the group discussions preceded the interviews and they recognised you from the in-depth discussions about the importance of different diseases, could this have affected the way the responded?

In –depth interviews occurred first. The interviewees were invited to attend meeting group A. M&Ms have been altered to convey this to the reader.

Line 474: As they did not normally have access to a veterinarian, how did you introduce yourselves in the discussions, could the mentioning of a veterinary degree in one of the interviewers have affected the line of conversation? I think this is what you are hinting at and I suggest making it more clear.

Hopefully the above changes have addressed this very valid point. The following statement has been added to the discussion:

“In the current study it is likely that participants were chose to focus on discussion of livestock disease syndromes as a consequence of the facilitators introducing team members as veterinary doctors.”

Line 478: Please specify the gender of the CKWs. I suggest a brief reflection on whether this could have mattered for the discussions.

The following statement has been added:

“The team included a female veterinary clinician, however it would have been preferable if female community workers had also been available, as this would assist in enabling women’s voices to be fully heard.”

Line 515: The positive quote indicates that the participants may have expected your listening to them to lead to more than a scientific paper, was this discussed or acted upon?

I’m glad you asked, This did indeed lead to more than a scientific paper. Farmer education initiatives a have subsequently been carried out based on the study’s findings. This is now mentioned in the materials and methods.

Lines 522-523: Were any official control measures prompted by the outbreaks and, if so, could this have affected the responses?

Good point, the following has been inserted:

“FMD outbreaks are common in the region, along with reactive vaccination programmes focused on the villages close to the KNP. Goat pox was however a novel disease in the villages (despite being classes as endemic in the region), and this resulted in a high degree of fear and uncertainty among farmers with respect to this condition (9). As a result of these investigations the presence of goat pox was reported to the local authorities and a vaccination programme initiated.”

Line 539: I suggest clarifying that female farmers are more likely to be the ones caring for these animal species and hence more likely to discuss them.

“Women are frequently the main carers of poultry, young animals and small ruminants.” Has been inserted for clarity.

Line 540: Please remove s from comments (Grace et al – plural) Done

Line 546: couldn’t Ranikhet also be avian influenza?

Thank you for raising this point, Avian influenza is a very important differential and so I have added it as such to Table 1 in the results section, and added the following footnote to the table. I think this makes this clear, so I have not altered the discussion with respect to this point, unless you feel that this would also be advantageous.

“Ranikhet: Newcastle disease appears the most likely diagnosis as the primary signs were neurological, particularly “sleepiness”, dung was reported to be white in colour, and ocular signs were not reported by participants. However, Avian influenza is an important differential diagnosis for this condition.”

Line 550: Please remove “and” before “making them…” Done

Line 585: I suggest using the past tense (did instead of does), to avoid confusion, and to specify if you refer to the current study or a previous one (I assume that the reference is a previous study but this could be written more clearly) Done

Line 587: From other LMIC studies, it appears that fertility issues are not recognised as “diseases” or “problems” if they are common. Perhaps a comment on the different views on problems that occur out of the ordinary and are regarded as novel as compared to things that are more or less expected would be useful? 

Indeed, I would include fertility in an underestimation of the productive capabilities of animals, similar to the issues around the weight of goats. I have added the following statement:

This could indicate low expectations of the productive capabilities of animals as a result of farmers having long term experiences of suboptimal productivity, effectively normalising this situation.

Lines 598-599: I suggest inserting “themselves” between “identifying” and “as farmers” and “the” between “urbanisation of” and “population”. Done

Line 609: This is a very bold statement, are you sure the information is an accurate representation of the participants’ perceptions? I would prefer a more modest phrasing, as the perceptions of the participants were presented to you in a certain situation, and interpreted by you.

I have wound my neck in and removed the first sentence from this paragraph!

Line 621: Again, are you sure the data highlight opportunities for interventions, did the participants say they wanted this? Or do you mean that the data indicate that there is a potential for improvement and that interventions such as those you suggest could, if implemented, improve animal health etc.? I may have missed this but I haven’t seen any direct results on suggested interventions and how implementation of different interventions would be easy to achieve.

Fair point. I have changed highlight to indicate. Participants directly requested interventions, and I have testimonials that could be inserted. A number of interventions have occurred as a result of this work, however we are still collecting data on this.

I have inserted the following two testimonials earlier in the discussion:

“Participants were very keen to learn more about animal health and the treatment of disease, making themselves both more self- sufficient and more productive. In the words of one male farmer: “I would like to learn from a Doctor or compounder (paravet) how to treat my own animals… Practical teaching. Meetings would be very useful. I could learn something from leaflets also… No one ever teaches us… Because it is for me I will have to manage time, and I will manage.” Participants displayed a strong sense of responsibility and expressed firm desire to increase their animal heath skills. Another male farmer positively asserted: “I can do this for my own cattle and my community.” On the subject of animal health education”

References

Not all references appear to entirely follow the journal instructions, please check and revise where relevant.

Reviewer #2: The article describes a participatory rural appraisal conducted in Kaziranga region, Assam state, India, to identify the main animal health issues affecting smallholder livestock farmers and explore their impact.

While the study itself is interesting, the manuscript, as it stands now, is too descriptive to qualify as a scientific article.

While there are a great many common global challenges, regional nuances of challenge, geography and socio- economic factors are of huge importance. This manuscript is built on rich data collected using social science techniques, and the use of this data to properly understand the local situation is key to the future design of effective interventions. While an unusual article, we believe that the descriptive nature of the article is part of its strength. 

 Based on the methods and themes raised by the authors in the interviews (lines 161-170) more results and discussion are expected on the impact of animal health issues, and natural hazards like flood, on farmers’ wellbeing, distinguished by their different effects, e.g. on nutrition, income, disturbance of farming activities and stress, and farmers’ attitude towards the risks. No such results are displayed and the authors do not explain the added value of their method compared to more classical impact assessments (stated in lines 614-615). Instead, they only list the health challenges recognized as “important” by farmers, but without explaining what this importance is based on. 

Thank you for raising this important point for clarification. Testimonials in the results section “Farming, environmental challenges, conflict with wildlife, and seasonal flooding” (Line 311- 381) capture aspects of the wide reaching impact of the seasonal flood and human- wildlife conflict on farmers. Additional material has been added to this section, as detailed below:

All 18 interviewed households discussed the seasonal flood, and three households stated that it was the greatest barrier to agricultural productivity.

Lack of sufficient fodder for livestock was raised as a major barrier to productivity by six households.

“Less availability of grass… I am unable to buy good quality food for my animals. We don’t have that kind of money so we are unable to provide good nutrition. Sometimes I buy food, but not enough for all my animals, and this reduces production. Husk and grain that I mix in cooked food… Animal nutrition is the biggest problem from March to July.”

Interviewee 11. Male, 58

The problem of animal fodder becomes particularly acute from the later part of the dry season as plant growth slows, and then during the seasonal flood grazing land is covered, and providing adequate nutrition for livestock at this time is especially problematic and a limit to productivity.

14 households raised the issue of crop raiding.

 “We mostly rely on crops. Crop raiding by elephants and wild boar is our most important problem. This is the main problem and nothing else. Sometimes elephants eat banana trees. It is in their nature, it is not a big problem. Forest guards do come to help us with thus problem. TCF have helped us providing tungi and torches. These have helped us to scare off elephants, we are very grateful to them.”

Interviewee 17. Male, 55

Eight interviewees discussed the problem of predation of livestock, six of whom related incidents when their animals had been killed. Two interviewees discussed predation as a major worry, but stated that their own animals had not been killed. 

Eight interviewees discussed wanting to purchase more highly bred animals, indicating that they consider the genetic potential of their livestock to be a limit to production. Four interviewees linked the acquisition of higher yielding animals with the need to secure more fodder resources in order to feed these animals to meet their potential.

“(I would) buy high quality hybrid cows to increase production. I think I can produce for most of the year, but there will not be sufficient fodder in flood time. That is why I am not buying (cows).”

Interviewee 18. Female, 25.

Two interviewees specifically mentioned money as a barrier to farm improvement, however 8 more suggested changes that they are currently unable to make which would require capital investment. Farmers perceive the barriers to farm improvement as complex and interlinked, with an interplay of lack of financial resources, wildlife conflict, and the seasonal flood.

“Because of flood, this ground is not high enough. I want to buy improved quality cows but because of the flood problem I am not buying. I want to make this ground higher, then perhaps I will buy. I want to make a good shed for the cows and then I can buy good quality cows and make more milk. I want to make a good shed with some metals (steel mesh) and then I can protect them from the tiger.”

Interviewee 11. Male, 58.

The following qualifying statements have been added in the discussion:

“In the current study it is likely that participants were chose to focus on discussion of livestock disease syndromes as a consequence of the facilitators introducing team members as veterinary doctors.”

“Formulating a list of farmers’ animal health concerns is not the same as assessing the true prevalence of disease or financial cost through classical epidemiological or economic techniques.”

In the materials and methods section, importance, for the purpose of this study, is defined as:

“take into account the severity of outcome, frequency of occurrence, duration of the problem, financial cost (including lost production, cost of medicines, and purchase of replacement animals), disruption to farming activities (including time of household members to care for animals or fetch fodder for those unable to graze and the loss of draught power), effect on human household nutrition, and emotional effect on the household.”

In retrospect it would have been preferable if another activity had been carried out to further elucidate this, such as participatory risk mapping. However, the testimonials and disease descriptions found in the supplementary materials provide significant insight into the effect of these challenges on local farmers.

The following relevant statements are also found within the discussion;

“An agricultural system which is both dependent on, and at risk from, a seasonal flood, is extremely vulnerable to the effects of the climate crisis (8). In addition to this, the loss of natural habitats, and changing patterns of rainfall, is resulting in increasing human- wildlife conflict in the region. In response, the farmers of the future in this region will have to be more adaptable in order to run productive units, however with the limited resources currently available this could be challenging to achieve.”

Further evidence of the impact of different conditions is provided by the testimonials and summery of descriptions in the supplementary materials section, and appropriate reference to these have been inserted into the revised manuscript.

Similarly, the authors do not make any real discussion on policy recommendation. The section “mitigation strategies” (lines 621-626) is very short and does not provide any new ideas for promoting interventions aims at improving animal health in smallholder systems: what are the most appropriate mitigation strategies? How mitigation strategies could be best implemented? What communication strategies would be effective?

Thank you for raising this important point. The section on mitigation strategies has been expanded, and now reads as follows:

The data presented in this study indicate opportunities for impactful interventions such as planned animal management, immunisation programmes, and selective use of anti-parasitic drugs. These could help to improve animal health, profitability, and sustainable land use. Conditions considered of greatest importance by the participants include four specific infectious viral disease plagues; three parasitic diseases: liver fluke, sarcoptic mange and parasitic gastro- enteritis; and three primary clinical signs: diarrhoea, bloat, and neck or throat swelling. Currently in the region vaccination is usually carried out for FMD only, usually on an ad-hoc basis in response to outbreaks. Planned programmes of vaccination for foot and mouth disease, haemorrhagic septicaemia, goat pox and Newcastle disease would have the potential to yield great benefits, while biosecurity education could help to limit the introduction and spread of these diseases. Education on care of neonatal ruminants, and particularly adequate colostrum feeding, could help reduce the incidence of diarrhoea in very young animals. Grazing management education could assist in the control of flukes, gastro-intestinal nematodes, and some causes of diarrhoea and bloat, as well as leading to more efficient use of available grazing resources. The success of these interventions depends on proper engagement of the community; and initiatives must be adapted to local community needs and conditions. Participants indicated a preference for face to face practical training, but also acknowledged the benefits of printed education materials in the local language. Linking health interventions, such as immunisation, to impactful farmer education, including the equal engagement of women farmers increases the potential for wide-ranging gains by improving animal health and productivity

Additionally, some methodological points deserve more explanation :

- The authors should describe the purpose of this ranking exercise: is it in order to confirm the outcome of the proportional pilling exercise?

Yes, that is correct. The following sentence has been inserted into the M&Ms:

“in order to review the results of this activity with the participants, a comparative ranking exercise was carried out. “

- Proportional piling scores on the importance of animal health issues are presented as a critical result of the study. But the authors do not explain on which criteria the score is based: severity of the disease, frequency, production loss, disturbance of farm activities? How are these different effects accounted for and combined to compute the final score? This deserves more explanation.

“Participants were asked to take into account the severity of outcome, frequency of occurrence, duration of the problem, financial cost (including lost production, cost of medicines, and purchase of replacement animals), disruption to farming activities (including time of household members to care for animals or fetch fodder for those unable to graze and the loss of draught power), effect on human household nutrition, and emotional effect on the household.”

Participants were given the respect to arrive at their own conclusions based upon the above. The statements and testimonials are summarised in the table in supporting information. This table could be brought into the main body of the manuscript, however it would add considerable length to the manuscript.

- In-depth individual interviews: were the three villages part of the ten villages where the authors conducted the village group meetings? What is their reference (A - J?)?

The in- depth interviews were carried out prior to the participatory village meetings (stated in the results). Interviewees were draw from villages A, I and J (now stated in M&M). Interviweees were then invited to attend meeting A, the following has been inserted into the M&M:

“Interview participants were invited to attend village participatory meeting A. Meetings I and J, though in the same cluster of villages, were attended by different farmers.”

 How were they selected? Why only three villages? How were the households selected for in-depth interviews? 

The team visited a cluster of three villages to undertake interviews. No attempt was made to randomly select participants, instead the focus was on interviewing farmers who were willing and talkative.

“All participants were volunteers, introduced to the project by a friend or relative, and care was taken to include animal keepers from a variety of social levels, families, and income levels within the village, and to ensure a balance of gender and age.”

What was the method for conducting interviews (semi-structured with the help of a checklist? Unstructured?)? If a checklist was used, it should be available in an appendix.

The aide memoire for the semi-structured interviews has been added to the supplementary materials

- In the list of identified diseases and condition names with likely diagnoses, which ones were confirmed through clinical examination?

The discussion states: 

“FMD (Chaboka / Kurra phatta); Newcastle disease (Ranikhet); Goat pox (Sagolay bohonta); sarcoptic mange (Papora dhora); were not only eloquently and consistently described by participants, clinical cases were also examined by the team, leading to a high level of certainty of diagnosis.”

This could be moved to the results if more appropriate? 

- Regarding gender differences: the authors should indicate whether females and males were interviewed separately and discuss the interest of conducting separate interviews.

The following description of the interviews can be found in the results section:

“Eight interviews were with a single main respondent: 5 men and 3 women; nine interviews were with a pair of main respondents: 4 married couples, 2 daughters with their mothers, 2 sons with their mothers, and 1 pair of brothers. One of these interviews was with three main respondents, a married couple and a male neighbour representing a separate household. Additional family members or friends frequently joined interviews for a time.”

In the results section: for each piece of information given in this manuscript (e.g. “The size and physique of an ox were considered particularly important, especially the musculature of limbs”), it would be good to indicate how many collective and individual interviews the information was extracted from. The authors should provide precise figures whenever possible rather than vague terms like "often".

Thank you for raising this point, the following statements, and supplementary table have been added:

“A table showing household income streams of interviewees can be found in Supplementary materials.”

“Large body size, provided it was in proportion, was stated by fifteen out of eighteen interviewees as a good sign”

“Udder size, teat conformation and large milk veins were considered positive factors for milk yield and ease of milking, mentioned by ten interviewees”

“Six interviewees mentioned aesthetic considerations such as colour or facial features, however four of these interviewees stressed that these were of secondary importance only.”

Other minor comments:

- Lines 469-474: do you have any figure showing the importance of animal health problems for farmers? If yes, include it in the result. Otherwise remove this sentence because it is not supported by data.

Data from the piling exercise only addresses animal health issues. The paragraph you refer to now reads (insertions in bold):

Meeting groups and interviewees tended to discuss disease syndromes affecting their livestock, rather than issues of grazing and fodder. The effects of the seasonal flood were often discussed, and were considered to be an immutable feature of life that had to be lived with, rather than a problem to be overcome. Crop raiding, and other conflict with wild animals, was also frequently discussed, but participants chose not to include these issues when asked “… to discuss the problems limit how much your animals contribute to your household” Similar to a study conducted in Himachal Pradesh (19) participants prioritised the discussion of animal health problems and infectious diseases, at least in part because they did not normally have access to a veterinary clinician to discuss these matters with. In the current study it is likely that participants were more likely to discuss livestock disease syndromes as a consequence of the facilitators introducing team members as veterinary doctors.

- On lines 477-487, the authors should avoid terms with connote

I am not wholly sure what connote are, however I have altered the section to ensure that all adjectives describing the activities describe only what could be directly observed (and not inferred) by the investigators. I hope that the revised passage, copied below, meets with your approval:

“Not all participants presented themselves equally in meetings, however facilitation by experienced local community workers from The Corbett Foundation created the space for all participants to be heard, and sought to minimise bias resulting from intra-community factors of social status, relative affluence, and gender. The team included a female veterinary clinician, however it would have been preferable if female community workers had also been available, as this would assist in enabling women’s voices to be fully heard.

Farmers contributed eagerly to open discussions on animal health, sharing their experiences and concerns. In common with the experiences of Ayele et al (21), Wright and Thrusfield (19), and Catley et al (22) proportional piling exercises were undertaken enthusiastically by participants, with good-natured debate as to the relative importance of different challenges. The discussion of these results and re-piling or re-ranking that followed resulted in lively discourse between neighbours before ultimately arriving at a consensus.”

Reviewer #3: This study is interesting and contributes to understanding the livelihoods, environmental challenges and livestock diseases of the small owners who live in the vicinity of a national park. Scientifically, it is a subject that deserves to be studied, however, the manuscript needs a lot of work to achieve a suitable version to be published in Plos-one.

 It is a difficult manuscript to evaluate, although the English language seems correct, the organization of the manuscript is poor and, in some cases, chaotic. 

Although the data is descriptive and qualitative in nature, the authors should explain how they analyzed it based on the objectives of the study, and not on the methodological tool. 

It is not clear how the information obtained from the various techniques used was combined. In the figures the authors have not translated into English the diseases described by the respondents. 

This information is contained in the manuscript and supplementary materials, which have been clarified in this revision. The table with the likely clinical diagnoses has now been moved forwards in the manuscript to precede the figures, however the local terms are still used, as per the work of Thrusfield and others, as local terms may encompass more than one condition, or more than one term be used for a number of clinical diagnoses, and as a result moving from local terms to English may not fully capture the situation. I hope that moving the table forwards in the manuscript aids the reader in properly understanding the figures.

The bibliography is scarce and does not cover the basic concepts on which the study is based. Authors should make an effort to find adequate bibliography.

The bibliography has been expanded in the revised version, it contains references detailing the techniques used, the challenges and opportunities faced by smallholder farmers, and relevant local information. I hope that you now find it satisfactory.

Abstract

The summary is discursive and does not correctly inform about the aim of the study, the methodology used, particulars of the participants and other characteristics of the study that allow the reader to know the real value of the study. Authors should develop a summary that is autonomous enough from the manuscript to understand the nature and value of the study.

Thank you for your suggestions. The revised abstract now reads as follows:

Improvements to smallholder farming are essential to improvements in rural prosperity. Small farmers in the Kaziranga region of Assam operate mixed farming enterprises in a resource limited environment, which is subject to seasonal flooding. Participatory techniques, were used to elucidate the animal health challenges experienced in this landscape in order to inform and guide future animal health education and interventions. 

The flooding is essential for agricultural activities, but is a source of major losses and disruption. Farmers experience significant losses to their crops due to raiding by wild species such as elephants; predation of livestock by wild carnivores is also of concern. Access to veterinary services and medicines is limited by both financial and geographic constraints. Interviewees discussed nutritional and management issues such as poor availability of fodder and grazing land, while meeting attendees preferred to concentrate discussions on animal health issues. Livestock keepers were adept and consistent at describing disease syndromes. The key challenges identified by farmers were: foot-and-mouth disease; Newcastle disease; haemorrhagic septicaemia; chronic fasciolosis; diarrhoea; bloating diseases; goat pox; and sarcoptic mange. Improvements in the efficiency of farming in this region is a prerequisite for the local achievement of United Nations Sustainable development goals. There exist clear opportunities to increase productivity and prosperity among farmers in this region through a combination of vaccination programmes and planned animal management schemes, driven by a programme of participatory farmer education.

Introduction

The authors dedicate the first four paragraphs of the introduction to the particular problems of the study area. This start gives a limited and local character to the study. Only the fifth paragraph reviews some generic farm animal health topics. The authors should rewrite this section starting with the problems that small producers generally face: 1) climate change or climatic adversities; 2) the socio-environmental conflicts and livelihoods of the populations that live near or are immersed in the national parks (i.e. Estevez-Moreno et al 2019 small ruminant research); 3) animal health and production under the conditions of smallholders, and 4) the importance of smallholder perceptions for future improvements in public policies. Later, they can explain the particularities of the study area, along with the justification, hypothesis and objectives of the study. This order will improve the reader's understanding of the conceptual framework of the study and the applicability of the study to the understanding of rural life not only in India but also in other similar agro-ecosystems in the world.

Thank you for bringing this to our attention. The following paragraphs has been added to the start of the introduction to give perspective to the importance of smallholders in global food security and achievement of UN SDGs, and the challenges and opportunities presented to these farmers:

“Smallholder farmers play an important role in global food production and engagement with them is an essential step in meeting United Nations Sustainable Development Goals. More than 80% of the world’ s poor people live in rural regions, and the majority of them are smallholder farmers (1). Small farms provide food, employment, community, and sustainable land use, benefitting poor people in rural regions all over the world. Farms under two hectares are estimated to provide 30- 34% of the world’s food, while utilising only 24% of the world agricultural land, and as farms increase in size local biodiversity falls, and post- harvest food losses increase (2). However, these farmers are often poor, and at risk of malnutrition, food insecurity and greater impoverishment (1). 

Numerous factors can challenge the ability of smallholder farmers to run productive, profitable enterprises, these include: limited access to credit, seeds, markets and sufficient water; poor rural roads; and the impact of the climate crisis. At the same time there are great opportunities for development including: diversification of produce, empowerment of women, provision of rural employment, and improvements in prosperity and food security. Increased productivity in small scale livestock farming improves childhood nutrition. Increased agricultural productivity provides greater employment opportunities for landless labourers, as farmers move from undertaking daily labour themselves to employing labourers on their own holdings (3). Small holder farming utilises very efficient value chains. Increased livestock farming activities by rural small holders, if supported through knowledge transfer, advisory services and co-operatives, lead to more widely increased economic activity in rural areas, better nutrition for both rural and urban dwellers, and where women are able to access these services equally, the empowerment of rural women, which in turn leads to better outcomes for children (4).”

In addition, numerous other changes have been made in accordance with the suggestions of reviewers 1 and 2.

M&M

L116-133 The study area is described descriptively and in a colloquial style. The authors do not give accurate information about the geographical conditions, the climatic categories (i.e. Köpen classification), topographic and other information that allows the reader to understand the study environment. All three maps have poor resolution and are not very helpful in understanding the study environment.

The following statement has been added;

“Assam is a humid sub-tropical monsoon region (Kopen climate classification Cwa)”

L133 Authors should develop a paragraph explaining the study design that includes all stages and the sequence for all participatory techniques. There is no sociodemographic information that allows us to understand the population studied.

Sociodemographic information about the population is elucidated by the study, and is described in the results section.

141-143 I do not understand the usefulness of individually identifying each member of the team. 

These identifiers have been removed

L244-260 These are not results, this paragraph must be in M&M.

As the demographic details of the interviewees was unplanned, and their inclusion was a result of the evolution of the study process, I have considered this as a result (ie who these people were, their gender, age, family, occupation etc). I realise that this is a matter of opinion whether to include this information as a result or method.

The above points cover many of the alterations to the manuscript, but are not an exhaustive account of the further work done.

We would like to thank the reviewers again for their assistance and the time they have taken to provide constructive feedback to improve the manuscript. We hope that following these changes the manuscript now meets with your approval and is suitable for publication in PLoS ONE.

Yours faithfully

Andy Hopker, Naveen Pandey, Sophie Hopker, Rebecca Marsland and Neil Sargison

---

## [Editor Report · Decision Letter 1]

6 Aug 2020

Animal health perceptions and challenges among smallholder farmers around Kaziranga National Park, Assam, India: a study using participatory epidemiological techniques.

PONE-D-20-12276R1

Dear Dr. Hopker,

We’re pleased to inform you that your manuscript has been judged scientifically suitable for publication and will be formally accepted for publication once it meets all outstanding technical requirements.

Kind regards,

Simon Clegg, PhD

Academic Editor

PLOS ONE

Additional Editor Comments:

Many thanks for resubmitting your manuscript to PLOS One

Unfortunately the two original reviewers were unavailable. Therefore, rather than invite new reviewers, I have reviewed the manuscript myself, and as all comments have been addressed and the manuscript reads well, I have recommended it for publication

You should hear from the Editorial Office soon

It was a pleasure working with you and I wish you all the best for your future research

Hope you are keeping safe and well during these difficult times

Thanks

Simon

---

## [Editor Report · Acceptance letter]

14 Sep 2020

PONE-D-20-12276R1 

Animal health perceptions and challenges among smallholder farmers around Kaziranga National Park, Assam, India: a study using participatory epidemiological techniques. 

Dear Dr. Hopker:

I'm pleased to inform you that your manuscript has been deemed suitable for publication in PLOS ONE. Congratulations! Your manuscript is now with our production department. 

Kind regards, 

on behalf of

Dr. Simon Clegg 

Academic Editor

PLOS ONE